# A delayed fractionated dose RTS,S AS01 vaccine regimen mediates protection via improved T follicular helper and B cell responses

**Suresh Pallikkuth[1], Sidhartha Chaudhury[2†], Pinyi Lu[2,3], Li Pan[1], Erik Jongert[4], Ulrike Wille-Reece[5], Savita Pahwa[1]\***

[1]Department of Microbiology and Immunology, University of Miami Miller School of Medicine, Miami, United States; [2]Biotechnology HPC Software Applications Institute, Telemedicine and Advanced Technology Research Center, U.S. Army Medical Research and Development Command, Fort Detrick, United States; [3]Henry M. Jackson Foundation for the Advancement of Military Medicine, Rockville, United States; [4]GSK Vaccine, Rixensart, Belgium; [5]PATH's Malaria Vaccine Initiative, Washington DC, United States

**Abstract** Malaria-071, a controlled human malaria infection trial, demonstrated that administration of three doses of RTS,S/AS01 malaria vaccine given at one-month intervals was inferior to a delayed fractional dose (DFD) schedule (62.5% vs 86.7% protection, respectively). To investigate the underlying immunologic mechanism, we analyzed the B and T peripheral follicular helper cell (pTfh) responses. Here, we show that protection in both study arms was associated with early induction of functional IL-21-secreting circumsporozoite (CSP)-specific pTfh cells, together with induction of CSP-specific memory B cell responses after the second dose that persisted after the third dose. Data integration of key immunologic measures identified a subset of non-protected individuals in the standard (STD) vaccine arm who lost prior protective B cell responses after receiving the third vaccine dose. We conclude that the DFD regimen favors persistence of functional B cells after the third dose.

**\*For correspondence:**
spahwa@med.miami.edu

**Present address:** [†]Center for Enabling Capabilities, Walter Reed Army Institute of Research, Silver Spring, United States

## Introduction

Malaria is a communicable vector-borne disease caused by *Plasmodium falciparum*, a protozoan parasite that is transmitted to humans via *Anopheles* mosquitoes. The reported case incidence and associated deaths have declined globally over the years, but malaria is still a major threat to communities in affected areas. In 2017, malaria cases were reported to occur in 87 countries with an estimated 435,000 deaths of children under the age of five (*WHO, 2018*). Development of a protective vaccine has been challenging and currently there is no licensed malaria vaccine. Among vaccines in development, the RTS,S/AS01 vaccine is the most advanced and is now part of a large-scale pilot implementation program in children in selected African countries. This vaccine includes three main components: a) portions of the circumsporozoite protein (CSP) of *Plasmodium falciparum*, which is the primary surface antigen (Ag) on the sporozoites; b) hepatitis B surface antigen (HB); and c) a proprietary AS01B adjuvant from GlaxoSmithKline (GSK). The adjuvant is composed of 3-O-desacyl-4'-monophosphoryl lipid A (MPL) from *Salmonella minnesota* and a saponin molecule (QS-21) purified from an extract from the plant *Quillaja saponaria*, combined in a liposomal formulation consisting of dioleoyl phosphatidylcholine and cholesterol in phosphate-buffered saline solution (*Vekemans et al., 2009*). A vaccine delivery regimen in which three doses of RTS,S/AS01B are given

on a 0, 1, 2 month schedule was found to provide nearly 50% protection in controlled human malaria infection (CHMI) trials (*Kester et al., 2009*; *Ockenhouse et al., 2015*). In the field, however, a Phase 3 trial of RTS,S/AS01 vaccine using the same dosing schedule showed only modest efficacy of 30.1% in 6–12-week-old infants (*Agnandji et al., 2012*).

In search of better efficacy, a recent CHMI trial of RTS,S/AS01 vaccine (Malaria-071, NCT01857869) was designed to compare the standard (STD) 0, 1, 2 month dosing regimen with a delayed fractional dose (DFD) regimen in which the third dose was one fifth of the standard dose and was administered at 7 months after the second dose (*Regules et al., 2016*). The rationale for testing the DFD regimen was based on a prior study from 1997 that had found that delaying and reducing the third immunization dose enhanced the immunogenicity of RTS,S vaccine and achieved efficacy to 86% (*Stoute et al., 1997*). Interestingly, Malaria-071 confirmed the earlier findings and the DFD regimen again showed 86% efficacy, which was significantly greater than the 62.5% protection observed in the STD regimen. To understand the factors associated with protection against *Plasmodium* infection, a number of immunologic investigations were conducted in a blinded manner.

The present study focused on assessing the role of peripheral T follicular helper (pTfh) cells and B cells, because a prior CHMI trial of the RTS,S vaccine had found an association of anti-CSP antibody titers and CSP-specific CD4 T cells with protection (*White et al., 2013*). In Malaria-071, antibodies of higher avidity were elicited in the DFD regimen in association with higher somatic hypermutation of B cells, suggesting fundamental changes in the maturation of B cell affinity (*Regules et al., 2016*). High-affinity antibodies are generated from long-lived plasma cells and memory B cells that are produced after antigen-primed B cells undergo cognate interaction with T follicular helper cells (Tfh). This interaction occurs in the germinal centers (GC) of secondary lymphoid organs (reviewed in *Crotty, 2011*), causing the B cells to proliferate followed by isotype switching and somatic hypermutation. Many properties of lymphoid Tfh cells, including B cell helper function for antibody (Ab) generation (*Crotty, 2011*), are also present in a subset of circulating CD4 T cells designated as pTfh cells that are considered as having emigrated from the lymphoid pool into the peripheral circulation (*Vella et al., 2019*; *Pahwa, 2019*). To investigate their role in human vaccine trials, pTfh in circulation serve as an attractive alternative to lymphoid Tfh, which require lymph node biopsies (*Bentebibel et al., 2016*; *Bentebibel et al., 2013*; *Herati et al., 2017*; *Herati et al., 2014*; *Pallikkuth et al., 2017*; *Pallikkuth et al., 2012*; *Pallikkuth et al., 2019*; *Boswell et al., 2014*; *Cubas et al., 2013*; *Locci et al., 2013*; *Simpson et al., 2010*; *Ueno, 2016*).

Only a few studies of pTfh have been performed in the context of immunogenicity and the efficacy of malaria vaccines. In a murine model, a nanoparticle-based vaccine presenting recombinant *P. vivax* CSP led to a protective immune response, characterized by enhanced GC formation with expansion and differentiation of antigen-specific Tfh cells (*Moon et al., 2012*). A malaria vaccine study in humans involving RTS,S/AS01 alone or co-administered with different viral-vectored vaccines showed that skewing of pTfh cells towards a CXC chemokine receptor 3 (CXCR3[+]) Th1 phenotype was associated with reduced Ab quantity and quality and lower vaccine efficacy (*Bowyer et al., 2018*). More recently, in a phase III trial of the GSK malaria vaccine 'Mosquirix' in Tanzania and Mozambique, children with increased frequencies of pTfh and plasmablasts at the time of vaccination exhibited higher Ab titers (*Hill et al., 2020*). An important role was ascribed to antigen-specific pTfh and their cytokine profile in influenza vaccine-induced antibody responses (*Pallikkuth et al., 2019*). In the present study, investigation of the dynamics of CSP-specific pTfh and B cell responses in the DFD and STD regimens of Malaria-071 (*Regules et al., 2016*), pre- and post-vaccination, revealed key immune features that were linked with protection after sporozoite challenge and provided insight into the superiority of the DFD regimen.

## Results

### CSP-specific pTfh responses are elevated in protected subjects

A scheme outlining vaccine timepoints and blood-sample collection for the immunological analyses is shown in *Figure 1*. Samples were analyzed at eight different timepoints, designated T0-T7: pre-vaccination (T0), day 6 post dose 1 (T1), day 28 post dose 1 (T2), day 6 post dose 2 (T3), day 28 post dose 2 (T4), day 6 post dose 3 (T5), day 21 post dose 3, pre-challenge (T6) and at study end, 159

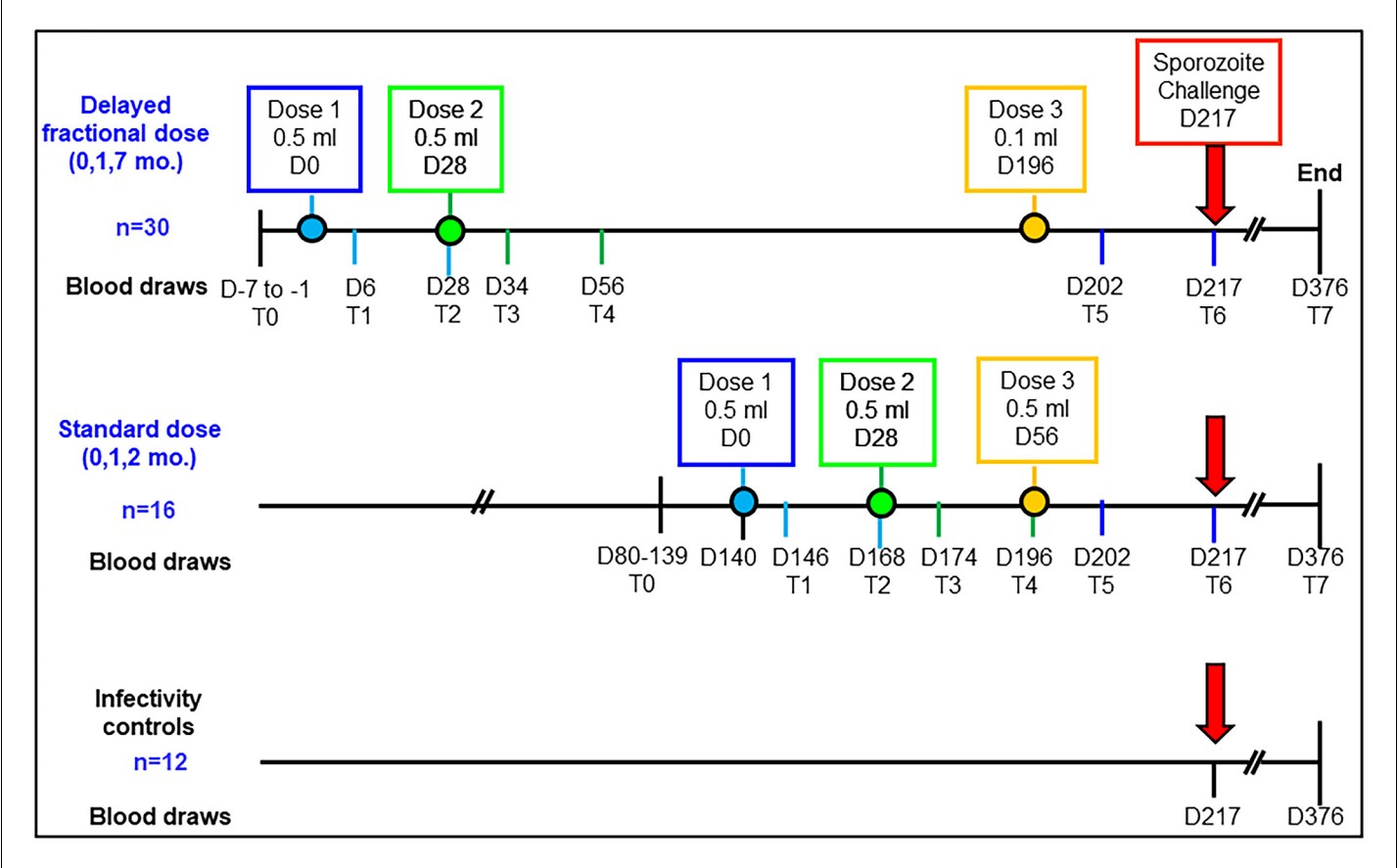

**Figure 1.** Study schema and assay timepoints. Timings of the first, second and third vaccine doses in either the standard dose regimen or the delayed fractional dose regimen are depicted in blue, green and yellow circles, respectively. Blood draws for immunology studies were performed at 8 timepoints designated T0 to T7: pre-vaccination (T0), day 6 (T1) and day 28 post first vaccination (T2), day 6 post second vaccination (T3), day 28 post second vaccination (T4), day 6 post third vaccination (T5), day 21 post third vaccination (T6, day of challenge) and at study end (T7, day 376; 159 days post-challenge).

days post-challenge (T7). The timing of the blood draws on day 6 and day 28 after each vaccine dose was designed to capture important periods for pTfh cell and B cell development post immunization. The distribution of protected (P)/non-protected (NP) participants was 10/6 in the STD regimen, and 26/4 in the DFD regimen (*Regules et al., 2016*). Given the small number of NP, we pooled data from both study regimens for each vaccine-induced immune response to understand the basis for protection.

Here, we analyzed the quantity and quality of CD4 T cells and pTfh cells to understand their role in vaccine-induced protection after RTS,S/AS01 vaccination. Circulating pTfh cells provide a snapshot of Tfh at the lymphoid inductive sites. Studies in healthy adults have documented the importance of pTfh expansion in response to vaccines as well as in the context of various infectious diseases (*Bentebibel et al., 2016*; *Bentebibel et al., 2013*; *Herati et al., 2017*; *Herati et al., 2014*; *Pallikkuth et al., 2017*; *Pallikkuth et al., 2012*; *Pallikkuth et al., 2019*; *Boswell et al., 2014*; *Cubas et al., 2013*; *Locci et al., 2013*; *Simpson et al., 2010*; *Ueno, 2016*). Data describing the frequencies of CSP-specific pTfh, along with total pTfh and CSP-specific CD4 T cells, in relation to P and NP status and the two vaccination regimens are shown in *Figure 2*. Detailed gating strategies for the identification of CD4 T cell subsets by flow cytometry are shown in *Figure 2—figure supplement 1*. We used CXCR5 expression on memory (CD45RO$^+$CD27$^+$) CD4 T cells to identify total pTfh cells. Expression of CD40L, an activation-induced molecule, was used to determine CSP-specific CD4 T cells after 12 hr in-vitro stimulation of peripheral blood mononuclear cells (PBMC)

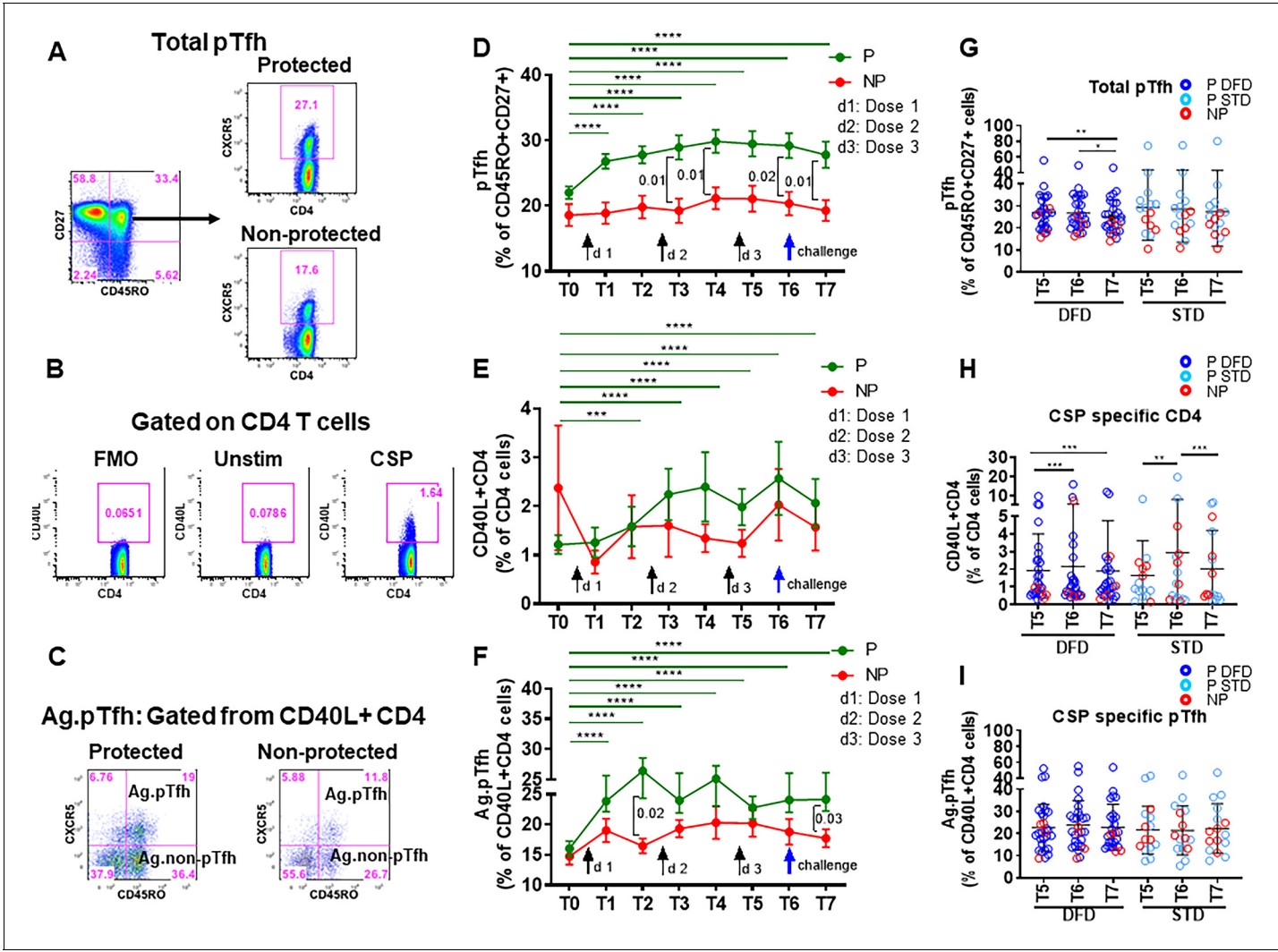

**Figure 2.** Higher frequencies of total pTfh and CSP-specific CD4 and CSP-specific pTfh cell responses in protected subjects. Frequencies of total pTfh, CSP-specific CD4 T cells and CSP-specific pTfh cells were identified by flow cytometry after 12 hr of PBMC stimulation with a CSP peptide pool in vaccinated subjects at different timepoints. Longitudinal data at different time points were analyzed for protected (P, n = 35) and non-protected (NP, n = 10) participants. (A–C) Flow cytometry dot plots for total pTfh cells, i.e. $CD45RO^+CD27^+CXCR5^+$ cells gated from CD4 T cells (A); CSP-specific CD4 T cells, i.e. $CD40L^+$ CD4 T cells (B); and CSP-specific pTfh cells, i.e. $CD45RO^+CXCR5^+$ cells gated from $CD40L^+$ CD4 T cells (C). (D–F) Line graphs with error bars indicating mean ± standard error of mean (SEM) for protected (green line) and non-protected (red line) individuals showing frequencies of total pTfh cells (D), $CD40L^+CD4$ T cells (E) and CSP-specific pTfh cells (F). (G–I) Scatter plots of CD4 T cell subsets in DFD and STD regimens at T5, T6 and T7 showing total pTfh cells (F), CSP-specific CD4 T cells (G) and CSP-specific pTfh cells (I) with data for the protected group represented by dark blue open circles for DFD (P DFD) and light blue open circles for the STD regimen (P STD), and the non-protected group represented by red open circles (NP) for both regimens. Statistical analysis was performed using the generalized linear mixed-effects model via Penalized Quasi-Likelihood to accommodate repeated measures over time. P values shown within the graphs refer to significant difference between the P and NP groups at the indicated time points. Statistical significance is shown as *p, <0.05; **, p<0.01; ***, p<0.001.

The online version of this article includes the following source data and figure supplement(s) for figure 2:

**Source data 1.** Total pTfh frequencies (*Figure 2D*).
**Source data 2.** Frequencies of CSP-specific CD4 T cells (*Figure 2E*).
**Source data 3.** Frequencies of CSP-specific pTfh cells (*Figure 2F*).
**Source data 4.** Frequencies of total pTfh: DFD vs STD (*Figure 2G*).
**Source data 5.** Frequencies of CSP-specific CD4 T cells: DFD vs STD (*Figure 2H*).
**Source data 6.** Frequencies of CSP-specific pTfh cells: DFD vs STD (*Figure 2I*).
**Figure supplement 1.** Gating strategy for the identification total pTfh, CSP-specific CD4 and CSP-specific pTfh.
**Figure supplement 2.** Frequency and function of CSP-specific non-pTfh did not differ between P and NP subjects.
**Figure supplement 2—source data 1.** Ag.non.pTfh frequency and function.

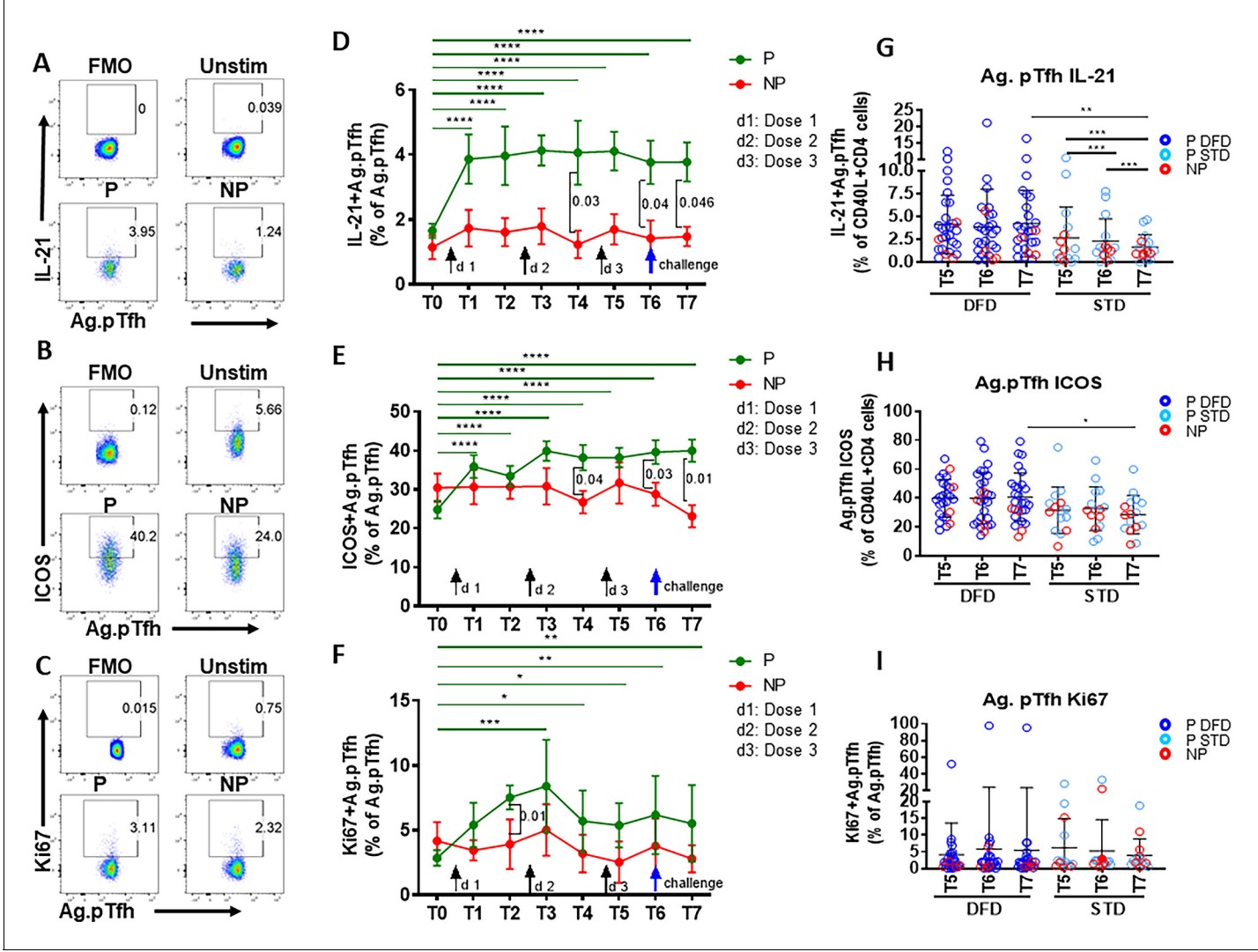

**Figure 3.** Higher induction of IL-21 and ICOS in CSP-specific pTfh cells from protected subjects. Representative flow cytometry dot plots showing (A) IL-21, (B) ICOS and (C) Ki67 expression in CSP-specific pTfh cells in protected (P, n = 35) and non-protected (NP, n = 10) subjects. (D–F) Line graphs with error bars indicate mean ± standard error of mean (SEM) for frequencies of IL-21$^+$ CSP-specific pTfh cells (D), ICOS$^+$ CSP-specific pTfh (E) and Ki67$^+$ CSP-specific pTfh (F) in the P (green line) and NP (red line) groups. (G–I) Scatter plots comparing IL-21$^+$ (G), ICOS$^+$(H) and Ki67$^+$ CSP-specific pTfh (I) in DFD and STD regimens at T5, T6 and T7. Data from the protected group are represented by dark blue open circles for DFD (P DFD) and by light blue open circles for the STD regimen (P STD), whereas data from the non-protected group are shown as red open circles (NP) for both regimens. Statistical analysis was performed using the generalized linear mixed-effects model via the Penalized Quasi-Likelihood to accommodate repeated measurements over of time. P values shown within the graph refer to significant difference between the P and NP groups at the indicated timepoints. Statistical significance is shown as *p, <0.05; **, p<0.01; ***, p<0.001.

The online version of this article includes the following source data for figure 3:

**Source data 1.** Frequencies of IL-21+ Ag.pTfh (*Figure 3D*).
**Source data 2.** Frequencies of ICOS+ Ag.pTfh (*Figure 3E*).
**Source data 3.** Frequencies of Ki67+Ag.pTfh (*Figure 3F*).
**Source data 4.** Frequencies of IL-21+Ag.pTfh: DFD vs STD (*Figure 3G*).
**Source data 5.** Frequencies of ICOS+ Ag.pTfh: DFD vs STD (*Figure 3H*).
**Source data 6.** Frequencies of Ki67+Ag.pTfh: DFD vs STD (*Figure 3I*).

with a CSP peptide pool. These CD40L$^+$ CD4 T cells were then gated for pTfh markers to determine antigen-specific pTfh (CD45RO$^+$CXCR5$^+$) andantigen-specific non-pTfh (CD45RO$^+$CXCR5$^-$) cells.

Representative dot plots from P and NP subjects for total pTfh, CSP-specific CD4 T cells, and CSP-specific pTfh are shown in *Figure 2A,B and C*, respectively. Frequencies of total pTfh were greater at all the timepoints post-vaccination than at T0 in P subjects, and these frequencies showed sustained expansion after vaccination compared to NP subjects at T3, T4 T6 and T7 (*Figure 2D*). Frequencies of CSP-specific CD4 T cells were significantly increased at T2–T7 compared to those at T0 in P subjects (*Figure 2E*). CSP-specific pTfh cells showed a strong vaccine-induced expansion in P subjects and were more numerous at all the timepoints post-vaccination than at T0 (*Figure 2F*). Importantly, neither total pTfh, CSP-specific CD4 nor CSP-specific pTfh cells showed an increase post-vaccination in NP subjects. At T5, T6 and T7, the late timepoints after the two regimens diverged, the frequencies of total pTfh, CSP-specific CD4 and CSP-specific pTfh did not show differences between the STD and DFD regimens (*Figure 2G, H and I*, respectively), but the frequencies of total pTfh and CSP-specific CD4 T cells in the DFD regimen showed a decline at T7, the last time point that was evaluated (*Figure 2G and H*). Frequencies of CSP-specific non-pTfh did not show an increase from T0 following vaccination and did not differ between the two groups or the two regimens (*Figure 2—figure supplement 2A and E*).

## IL-21$^+$ and ICOS$^+$ pTfh subsets are associated with protection

To investigate the quality of vaccine-induced CD4 T cells in the context of protection, we analyzed (i) CSP antigen-induced intracellular IL-21 (*Figure 3D*), the signature Tfh cytokine; (ii) expression of inducible co-stimulatory molecule (ICOS) (*Figure 3E*), which is associated with the follicular recruitment, maintenance and function of Tfh cells, and (iii) Ki67 (*Figure 3F*), a marker indicative of cellular activation and proliferation. A significant increase in the frequencies (compared to those at T0) of IL-21-expressing (*Figure 3G*) and ICOS$^+$ (*Figure 3H*) CSP-specific pTfh cells was evident in P subjects at all timepoints post vaccination and of Ki67$^+$ (*Figure 3I*) CSP-specific pTfh cells at T3–T7. When comparing CSP-specific pTfh of P to NP subjects, frequencies of IL-21$^+$ and ICOS$^+$ cells showed an increase at T4, T6 and T7 (*Figure 3D and E*) and a transient increase of Ki67$^+$ CSP-specific pTfh cells at T2 in P subjects (*Figure 3F*). In the NP subjects, no increase in IL-21$^+$ or ICOS$^+$ or in Ki67$^+$ CSP-specific pTfh was noted post-vaccination, and the levels remained at background levels (*Figure 3D, E and F*), with ICOS$^+$ cells dipping even lower at T7 (*Figure 3E*). ICOS$^+$ total pTfh (CD45RO$^+$CD27$^+$CXCR5$^+$ CD4 T cells) were present at higher frequencies in P subjects compared to baseline levels at T3-T7 and at higher frequencies than in NP subjects at T6 and T7 (not shown). IL-21$^+$ total pTfh also showed a trend for higher frequencies in P subjects compared to NP subjects post-vaccination (not shown). Frequencies of CSP-specific IL-21$^+$, ICOS$^+$ and Ki67$^+$ non-pTfh cells did not change post-vaccination and did not differ significantly between P and NP subjects at any timepoint (*Figure 2—figure supplement 2B, C and D*).

Comparing the two regimens, we noticed that IL-21- and ICOS-expressing CSP-specific pTfh were significantly more frequent at T7 in the DFD regimen, but did not show a difference in Ki67 expression (*Figure 3G, H and I*, respectively). In the STD regimen, the frequencies of IL-21$^+$ CSP-specific pTfh decreased at T7 from those at T5 and T6 (*Figure 3G*). In the CSP specific non-pTfh compartment, frequencies of IL-21$^+$, ICOS$^+$ and Ki67$^+$ non-pTfh cells did not differ between the DFD and STD regimens at T5, T6 or T7 (*Figure 2—figure supplement 2F, G and H*). Taken together, these data demonstrate that, as a group, P subjects show vaccination-induced expansion of both total and functional CSP-specific pTfh cells that respond to Ag stimulation with IL-21 production and ICOS and Ki67 expression, whereas NP subjects do not do so.

## CSP-responsive B cells emerge after the second dose in the P subjects and are more frequent in the DFD regimen

To test the impact of vaccination on the B cell compartment, we first analyzed alterations in B cell maturation subsets ex vivo. The gating strategy for B cell subsets by flow cytometry is shown in *Figure 4—figure supplement 1*. Total B cells were identified as CD3$^-$CD20$^+$ cells and total memory B cells as CD20$^+$CD27$^+$ cells. On the basis of the expression of CD21, CD27 and IgD, B cell maturation subsets were identified as naïve (CD21$^{hi}$IgD$^+$CD27$^-$), resting memory (RM, CD21$^{hi}$CD27$^+$), activated memory (AM, CD21$^{low}$CD27$^+$) and atypical memory B cells (aMBC, CD21$^{low}$CD27$^-$). Further, on the

basis of the expression of IgD and IgG, switch and unswitch memory B cell subsets were identified as total switch memory (SM), total unswitch memory (UM), switch RM (sRM), unswitch RM (uRM), switch AM (sAM), and unswitch AM (uAM) (*Figure 5—figure supplement 1*). Neither total B cells nor any of the ex-vivo-derived B cell maturation subsets differed significantly between P vs. NP subjects at any timepoint or between the two vaccine regimens at T5, T6 and T7 (not shown). The expression of CD80, a marker indicative of T-cell-dependent B cell activation, was analyzed ex vivo in total B cells, and in RM and AM subsets (*Figure 4—figure supplement 2A*), and did not differ significantly at any timepoint between P and NP subjects (*Figure 4—figure supplement 2B, D and F*, respectively), or between the DFD and STD regimens at the later timepoints (T5, T6 and T7) (*Figure 4—figure supplement 2C, E and G*, respectively). There was a trend for higher CD80 expression in AM B cells at T6 and T7 in P subjects (*Figure 4—figure supplement 2F*).

To assess the functional properties of B cells, we cultured PBMC with (i) full-length CSP protein (PF-CSP), (ii) the CS repeat region (R32LR) and (iii) the C-terminal peptide (PF-16) to clarify whether there was a region-specific dominant response to CSP in the B cell compartment. Examination of the B cell phenotype in the antigen-stimulated cultures was performed to assess memory B cell subsets by flow-cytometry and proliferation using Ki67. Changes in antigen-stimulated B cell phenotypes were noted mostly in relation to the regimen and not in the context of protection. Regimen-specific differences emerged post dose 3 (T5 or later) exclusively in the Ag-specific memory B cell compartment, including SM and sAM for PF-CSP (*Figure 4A and B*) and PF-16 (*Figure 5D and E*), with larger responses in the DFD arm compared to the STD arm. Ki67$^+$ B cells were also more frequent following PF-CSP and PF-16 antigen stimulation at T5 and T6 in the DFD arm (*Figure 5C and F*). Frequencies of PF-CSP-specific Ki67$^+$ aMBC B cells were also higher in the DFD arm at T5 and T6 (*Figure 4—figure supplement 3*). In the context of protection, although the levels of all subsets tended to be higher in P than in NP, none reached significance except for PF-16-stimulated Ki67$^+$ memory B cells, which increased from T0 to T6 (*Figure 4—figure supplement 4*).

Functional assessment of the vaccine-induced plasmablast and memory B cell antibody responses against the test antigens was conducted using antibody secreting cell (ASC) ELISpot assays. Plasmablasts are short-lived ASC that are generated rapidly in response to infection or vaccination, which transiently contribute to serum antibodies (*Wrammert et al., 2008*; *George et al., 2015*; *Pallikkuth et al., 2011a*; *Rinaldi et al., 2017*). To assess plasmablast responses, we determined the number of spontaneous IgG ASC directed at vaccine antigens on day 6 post-each vaccine dose as compared to the number pre-vaccination (*Figure 5*). A significant increase in the number of PF-CSP-specific (*Figure 5A*) and R32LR-specific (*Figure 5B*) spontaneous ASC were noted at day 6 post dose 2 (T3) in P subjects but not in NP subjects. The spontaneous ASC response did not differ significantly between the DFD and STD regimens at T5 (not shown).

Vaccine-induced antigen-specific IgG secreting memory B cell responses were analyzed at day 28 post-vaccination by memory B cell ELISpot assay following in vitro antigen stimulation (*George et al., 2015*; *Rinaldi et al., 2017*; *Pallikkuth et al., 2011b*). Memory B cells are mainly generated in the GC in secondary lymphoid organs. After leaving the GCs, memory B cells either join the recirculating pool of lymphocytes or home to antigen-draining sites. Kinetics, as well as the CSP epitope specificities of the vaccine-induced functional memory B cell responses, were analyzed (*Figure 5*). Comparing P to NP subjects, we found that only the P subjects showed an increase in memory B cell response to PF-CSP protein from T0 to T4, T6 and T7, and that at these time points, both the PF-CSP response and the response to the PF16 region were greater in P than in NP subjects (*Figure 5C and D*). The repeat region R32LR-specific memory B cell response also increased in P subjects only from T0 to T4 and T7, and was greater in the P group than in the NP group at T4 (*Figure 5E*). Comparing the regimens, in the DFD regimen the memory B cell response to PF-CSP increased from T5 to T6 and T7 (*Figure 5F*), and the response to PF-16 at T7 (*Figure 5G*) was larger under the DFD regimen than under the STD regimen.

As an additional measure of memory B cell function, we analyzed IgG secretion by ELISA in the PBMC culture supernatants after 5 days of stimulation with PF-CSP, PF-16, or R32LR antigens (*Figure 5—figure supplement 1*). Compared to T0, PF-16-specific IgG levels were significantly higher at T4, T6 and T7 in P subjects (*Figure 5—figure supplement 1A*), whereas PF-CSP- and R32LR-specific IgG did not change significantly (*Figure 5—figure supplement 1C and E*). IgG responses were significantly higher in the DFD regimen at T5–T7 for PF-16 (*Figure 5—figure supplement 1B*) and at T7 for R32LR (*Figure 5—figure supplement 1F*), with a trend of higher response at T7 for PF-CSP

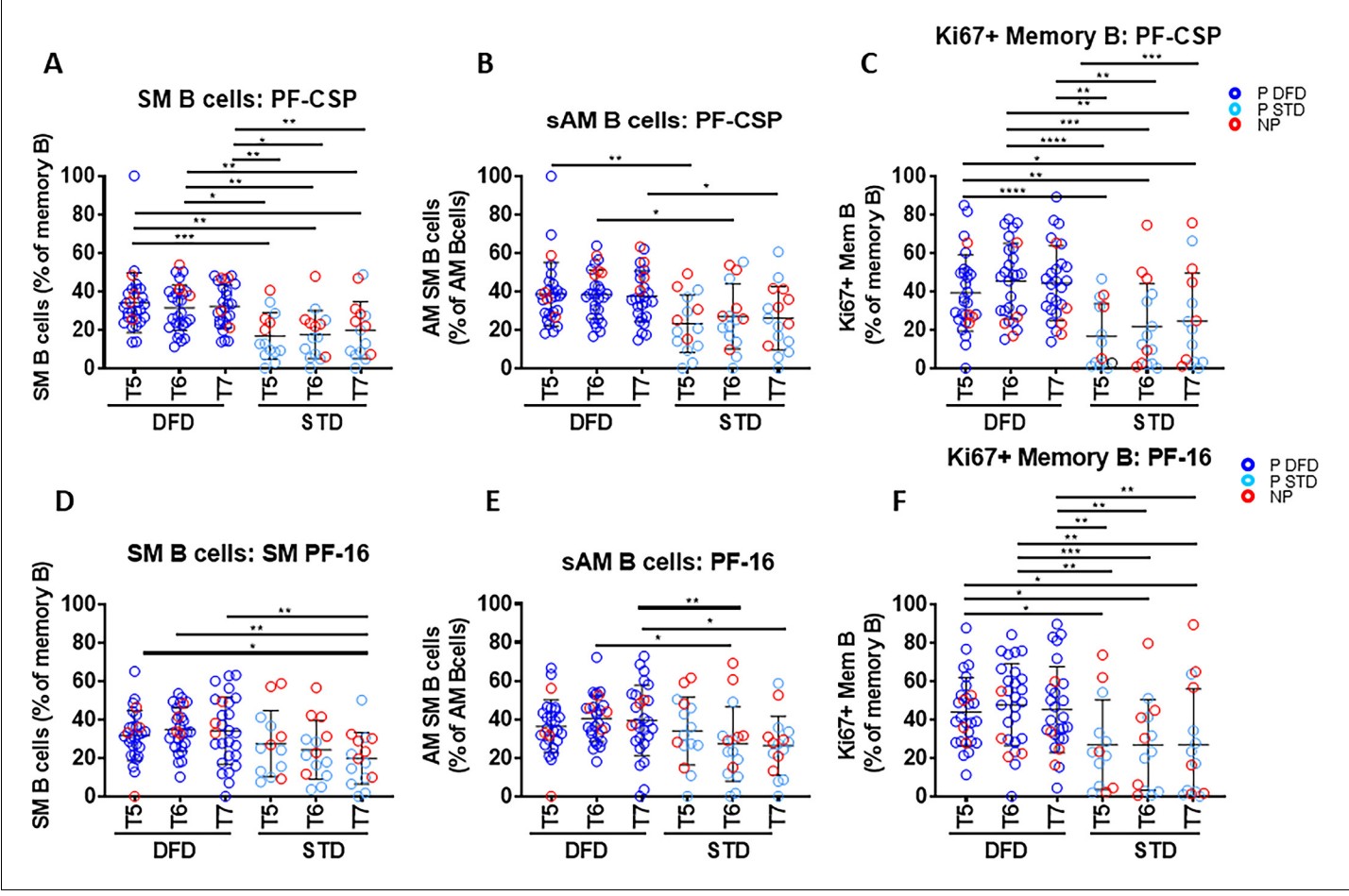

**Figure 4.** Frequencies of CSP-specific memory B cell subsets are greater in the DFD regimen than in the STD regimen at later time points. PBMC were cultured for 5 days in the presence of PF-CSP and PF-16 antigens and analyzed for frequencies of B cell maturation subsets: switched memory (SM: CD20$^+$CD10$^-$CD27$^+$IgD$^-$IgG$^+$), switched activated memory (sAM: CD20$^+$CD10$^-$CD21$^{low}$CD27$^+$IgD$^-$IgG$^+$) and Ki67 expression on total memory (Mem: CD20$^+$CD10$^-$CD27$^+$) B cells in DFD (n = 29) and STD (n = 14) regimens at T5, T6 and T7. (A–C) The scatter plots show PF-CSP-specific SM (A); sAM (B) and Ki67$^+$ (C) memory B cells. (D–F) PF-16-specific SM (D); sAM (E) and Ki67$^+$ (F) memory B cells. Data for the protected group are represented as dark blue open circles for DFD (P DFD, n = 25) and as light blue open circles for STD regimen (P STD, n = 10) and non-protected as red open circles (NP, n = 10) for both regimens. Statistical analysis was performed using the generalized linear mixed-effects model via Penalized Quasi-Likelihood to accommodate repeated measurements over time. Statistical significance is shown as: *, p<0.05; **, p<0.01; ***, p<0.001.

The online version of this article includes the following source data and figure supplement(s) for figure 4:

**Source data 1.** PF-CSP-specific SM B cells: DFD vs STD (*Figure 4A*).
**Source data 2.** PF16-specific SM B cells: DFD vs STD (*Figure 4D*).
**Source data 3.** PF CSP switched activated memory B cells: DFD vs STD (*Figure 4B*).
**Source data 4.** PF 16-specific switched activated memory B cells: DFD vs STD (*Figure 4E*).
**Source data 5.** PF CSP-specific Ki67+ memory B cells: DFD vs STD (*Figure 4C*).
**Source data 6.** PF 16-specific Ki67+ memory B cells: DFD vs STD (*Figure 4F*).
**Figure supplement 1.** Gating strategy for the B cell subsets.
**Figure supplement 2.** CD80 expression on the total B cell, RM and AM subsets.
**Figure supplement 2—source data 1.** CD80 expression on B cell subsets.
**Figure supplement 3.** Higher atypical memory B cells (aMBC) at T5 and T6 in the DFD group.
**Figure supplement 3—source data 1.** Ki67+ aMBC specific to PF-CSP and PF-16.
**Figure supplement 4.** CSP-specific memory B cell subsets in P and NP subjects.
**Figure supplement 4—source data 1.** Mean frequencies of mory B cell subsets between P and NP subjects.

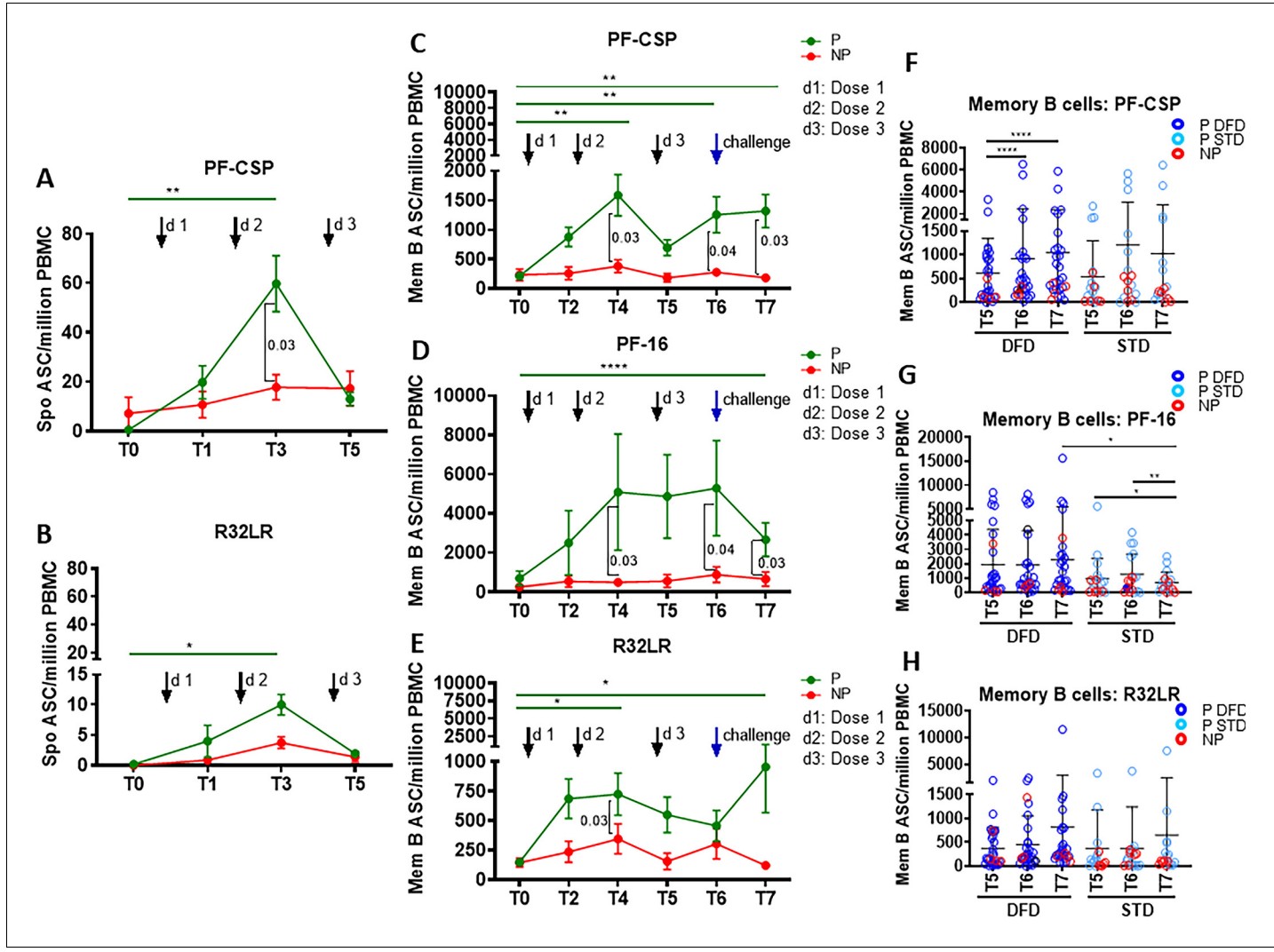

**Figure 5.** Higher CSP-specific plasmablast and memory B cell ASC responses in the P subjects. Spontaneous antibody secreting cells (ASC) at T0, T1, T3, and T5 were determined in unstimulated PBMC for plasmablast responses by ELISpot against PF-CSP, PF-16 and R32LR antigens. Memory B cell ASC responses were analyzed at T0, T2, T4, T5, T6 and T7 using ELISpot assay in PBMC stimulated with PF-CSP, PF-16, or R32LR antigens. Line graphs with error bars indicate mean ± standard error of mean (SEM). (A, B) Spontaneous ASC/million PBMC for the protected (P, green line, n = 33) and non-protected (NP, red line, n = 10) study groups for PF-CSP (A) and R32LR (B) antigens. (C–E) Memory B cell ASC/million PBMC for PF-CSP (C), PF-16 (D) and R32LR (E) for P and NP subjects. (F–H) Scatter plots showing memory B cells responses as ASC/million PBMC in the DFD and STD regimens at T5, T6 and T7 for PF-CSP (F), PF-16 (G) and R32LR (H) antigens. Data from the protected group are represented as dark blue open circles for DFD (P DFD) and as light blue open circles for the STD regimen (P STD), whereas data for the non-protected group are represented as red open circles (NP) for both regimens. Statistical analysis was performed using the generalized linear mixed-effects model via Penalized Quasi-Likelihood to accommodate repeated measurements over time. P values shown within the graph refer to significant difference between the P and NP groups at the indicated timepoints. Statistical significance shown as: *, p<0.05; **, p<0.01; ***, p<0.001.

The online version of this article includes the following source data and figure supplement(s) for figure 5:

**Source data 1.** Spontaneous ASC/million PBMC: PFCSP (*Figure 5A*).
**Source data 2.** Spontaneous ASC/million PBMC: R32LR (*Figure 5B*).
**Source data 3.** Memory B cell ELISpot: PFCSP (*Figure 5C*).
**Source data 4.** Memory B cell ELISpot: PF16 (*Figure 5D*).
**Source data 5.** Memory B cell ELISpot: R32LR (*Figure 5E*).
**Source data 6.** PF-CSP-specific memory B cell ELISpot: DFD vs STD (*Figure 5F*).
**Source data 7.** PF-16-specific memory B cell ELISpot: DFD vs STD (*Figure 5G*).
**Source data 8.** R32LR-specific Memory B cell ELISpot: DFD vs STD (*Figure 5H*).
**Figure supplement 1.** IgG specific to PF-16, PF-CSP and R32LR in PBMC culture supernatants.
**Figure supplement 1—source data 1.** IgG in culture supernatants.

(*Figure 5—figure supplement 1D*), in comparison to responses in the STD regimen. Taken together, our findings indicate that RTS,S/AS01 vaccination elicited strong, functionally competent CSP-specific memory B cell responses in the P subjects, especially at the later timepoints, and that these responses were larger in the DFD regimen and stronger for PF-16 than for R32LR.

## Data integration approach for identifying vaccine-induced immune correlates and their association with protection or regimen

In order to identify vaccine-induced immune correlates that are associated with protection and that differentiate the DFD regimen from the STD regimen, we employed a statistical data integration method. We incorporated data obtained for both CSP and HBs antigen-specific immune responses for this analysis, which include frequencies of memory B cell phenotypes, memory B cell ELISpot responses, CD4 and pTfh responses, and IgG levels from PBMC culture supernatants. We identified 676 of 1976 immune measures that were significantly increased from baseline (T0) to different timepoints post-vaccine (*Supplementary file 1*). By carrying out a correlation analysis to identify groups of correlated immune measures ('immune clusters'), we were able to group these 1976 immune features into 142 immune clusters, of which 40 clusters had at least one vaccine-antigen-specific immune feature. Analysis of the vaccine-induced responses over the time course of the study revealed that the pTfh response was an early-stage response, emerging as early as T2, and persisting throughout the study. By contrast, the memory B cell response was a later-stage response, peaking between T4 and T5 (*Figure 6—figure supplement 1A*). 65% to 80% of the immune responses classified as 'vaccine-induced' were specific to the vaccine antigens CSP or HBs (*Figure 6—figure supplement 1B*), and the response was fairly balanced between both antigens.

## Individualized predictions using machine learning

In order to assess the extent of regimen- and protection-level differences, we applied a machine-learning approach using random forest statistical modelling that could make individualized predictions of regimen and protection from immune data alone. A general workflow of the data integration approach is shown in (*Figure 6—figure supplement 2*). This approach allowed us 1) to determine what combination of immune features is most predictive of regimen or protection, and 2) to group subjects according to their pattern of vaccine-induced immune responses. Furthermore, by taking a prediction approach, we were able to determine how early in the vaccination regimen vaccine-induced immune responses would be predictive of protection. In order to assess predictive performance, we carried out a leave-one-out (LOO) analysis, in which each subject was excluded from the data set before the predictive model was trained on the remaining subjects, and then used to predict the outcome (or regimen) of that excluded subject. Accuracy was calculated as the proportion of subjects whose outcome (or regimen) was correctly predicted by the model.

In order to predict vaccine regimen from immune data alone, we performed a random forest analysis using 41 parameters from timepoints prior to challenge (T6) that were shown to be significantly different with respect to regimen in the univariate analysis. The LOO analysis shows that the random forest model, using these 41 parameters, achieved 85% accuracy with a kappa value of 0.63, indicating a strong predictive value. Overall, an average of 39 out of 46 subjects in the vaccine regimens were predicted correctly. Further, we determined the relative importance of each parameter in the random forest (*Table 1*) and found that antigen-induced B cell characteristics, including proliferation (Ki67$^+$) and frequencies of SM, sAM, and Ki67 expressing aMBC, were most predictive of regimen. Nearly all predictive parameters showed antigen specificity for either CSP (66%) or HBs (27%). We used principal components analysis (PCA) to visualize how well the predictive parameters identified in *Table 1* were able to distinguish subjects by regimen (*Figure 6A*). Overall, we found good separation between DFD and STD regimens using these parameters. We also found that the axis of variation within each regimen was distinct between the two groups, suggesting that these regimens are acting differently on this common set of immune parameters.

In order to predict protection status, we used 36 immune parameters that showed significant protection-level differences prior to challenge (T6 and earlier). We achieved a predictive accuracy of 85% with a kappa of 0.45, indicating low-to-moderate predictive ability, with 18 parameters in the model. Overall, 39 of 46 subjects were predicted correctly. The low sample size and imbalanced data set (78% of subjects were protected) made a more thorough assessment of the predictive

**Table 1.** Parameters that are most predictive of vaccine regimen.

| Cell type | Phenotype | Parameter | Weight |
|---|---|---|---|
| B cell | Ki67+memory B cells | BCF.Mem.Ki67.HBs.T5 | 100 |
| | | BCF.Mem.Ki67.PF.CSP.T6 | 73 |
| | | BCF.Mem.Ki67.MED.T6 | 55 |
| | | BCF.Mem.Ki67.PF.CSP.T5 | 53 |
| | | BCF.Mem.Ki67.PF.16.T6 | 45 |
| | sAM | BCF.sAM.PF.16.T6 | 91 |
| | | BCF.sAM.PF.CSP.T5 | 67 |
| | | BCF.sAM.HBs.T6 | 53 |
| | SM | BCF.SM.PF.CSP.T5 | 54 |
| | | BCF.SM.HBs.T5 | 50 |
| | aMBCKi67 | BCF. aMBC.Ki67.PF.CSP.T5 | 54 |
| | | BCF. aMBC.Ki67.PF.16.T5 | 48 |
| | | BCF. aMBC.Ki67.HBs.T5 | 44 |
| | | BCF. aMBC.Ki67.PF.CSP.T6 | 42 |
| | | BCF.sAM.Ki67.PF.CSP.T5 | 40 |

Abbreviations: sAM, switched activated memory; SM, switched memory; aMBCKi67, Ki67+atypical memory B cells; BCF, antigen-specific memory B cell responses by flow cytometry.

ability of this model challenging. After analyzing for variable importance, we found that the parameters that are most predictive of protection (*Table 2*) include CSP-specific CD40L+ CD4, HBs-specific IL-21+, CSP-specific pTfh, frequencies of total pTfh cells, and CSP-specific antibody-secreting memory B cells. Of note, many of these parameters were from relatively early timepoints

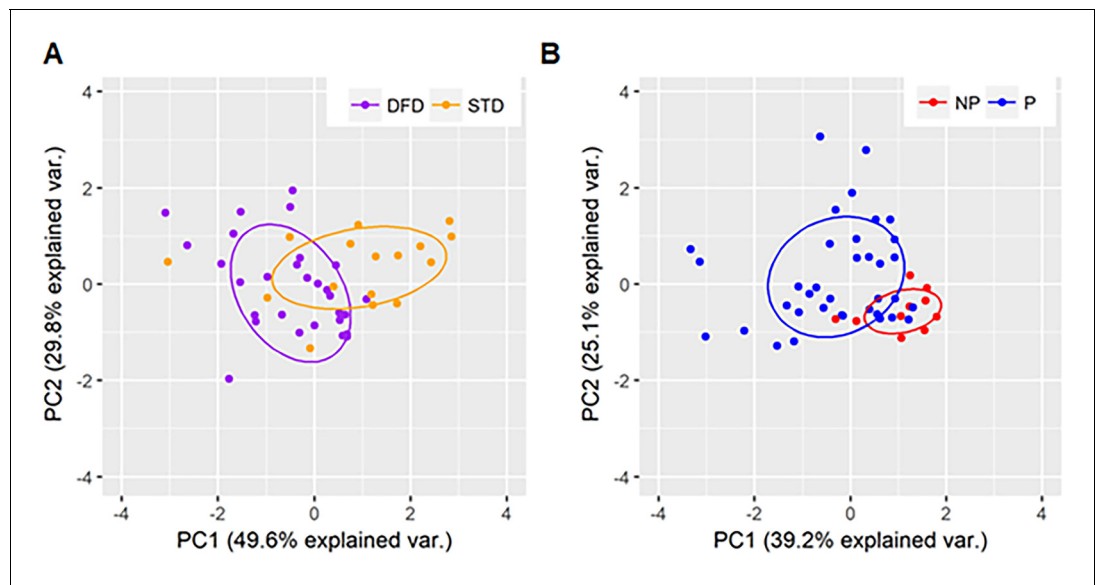

**Figure 6.** PCA plots showing regimen-specific and protection-status differences. PCA plots using parameters identified by machine learning as being predictive of (A) regimen differences for DFD (purple) and STD (orange) subjects and (B) protection status for protected (blue) and non-protected (red) subjects.

The online version of this article includes the following figure supplement(s) for figure 6:

**Figure supplement 1.** Summary of vaccine-induced responses.
**Figure supplement 2.** General workflow for the data integration approach.

**Table 2.** Parameters that are most predictive of protection.

| Cell type | Phenotype | Parameter | Weight |
|---|---|---|---|
| B cell | Memory B cell ELISpot | PF.CSP.T2 | 79 |
| | | PF.16.T4 | 64 |
| | | PF.CSP.T5 | 53 |
| T cell | CD40L$^+$CD4 | CD40L.CSP.T4 | 100 |
| | | CD40L.CSP.T6 | 57 |
| | | CD40L.CSP.T2 | 52 |
| | Ag.pTfh | Ag.pTfh.CSP.T2 | 71 |
| | | Ag.pTfh.IL.21.HBs.T4 | 60 |
| | | Ag.pTfh.IL.21.CSP.T6 | 56 |
| | | Ag.pTfh.IL.21.HBs.T6 | 55 |
| | | Ag.pTfh.IL.21.CSP.T4 | 49 |
| | Total. pTfh | Total.pTfh.MED.T2 | 49 |
| | | Total.pTfh.MED.T4 | 43 |
| | | Total.pTfh.MED.T6 | 40 |
| | Ag.CD4 | CSP.T4 | 46 |
| | | CSP.T6 | 38 |
| | | MED.T6 | 42 |
| | | HBs.T4 | 52 |

Abbreviations: Ag.pTfh, antigen specific peripheral T follicular helper cells; Ag.CD4, antigen stimulated total CD4 T cells; ELIspot, memory B cell ELISpot responses; CD40L$^+$CD4, CD40L+ CD4 T cells.

such as T2 and T4. We used PCA to visualize how well the predictive parameters identified in *Table 2* could distinguish subjects by protection status (*Figure 6B*), and found that although there was a wide variability in the immune responses for P subjects, NP subjects clustered closely with each other and separately from P subjects. Together, these data suggest that there is a distinct pattern of immune responses associated with vaccine failure in this study.

## Predicting protection from early-stage responses

Given that many of the immune correlates for protection were found at timepoints before dose 3, we used machine learning to determine whether we could predict if a subject could be protected by early-stage immune responses alone. We trained the model on early-response data alone (post-dose 1 and 2) to predict protection and achieved 87% accuracy with a kappa of 0.46, indicating moderate accuracy in predicting protection using only immune response data prior to dose 3. When we broke down the prediction results by vaccine regimen, we found that the protection status of virtually all DFD subjects is predicted correctly (97% accuracy, kappa = 0.84), whereas the protection status of STD subjects is predicted poorly (69% accuracy, Kappa = 0.26).

We stratified three classes of subjects: subjects whose early-stage immune responses were predictive of protection and who were actually protected, subjects whose early-stage immune responses were predictive of protection but who were not protected, and subjects whose early-stage immune responses predicted non-protection but who were, in fact, not protected. Interestingly, in both the STD regimen and the DFD regimen, approximately 15% of subjects (dark orange) elicited weak early-stage immune responses predictive of non-protection, and these subjects were subsequently found not to be protected following challenge.

In terms of subjects who elicited promising early-stage immune responses, we found that among DFD subjects, virtually all were in fact protected following dose 3 and challenge. By contrast, in the STD regimen, approximately one third of subjects with promising early-stage immune responses were not protected. These findings suggest that the third immunization in the STD regimen may adversely affect the immune response elicited by dose 1 and dose 2, and this may lead to the lack of protection.

On the basis of these individualized predictions of efficacy constructed on the early-stage immune response data, we were able to classify study participants into three groups of outcomes (*Figure 7A*): 1) 'weak responders', approximately 10–15% of subjects in both vaccine regimens (n = 6), who elicited poor early-stage immunogenicity and showed low efficacy (16% efficacy); 2) 'DFD strong responders', DFD subjects who showed promising early-stage responses (n = 27) and were almost entirely protected (96% efficacy); and 3) 'STD strong responders', STD subjects who showed strong early responses (n = 13) and achieved moderate protection (70% efficacy). To visualize these groups of outcomes, we generated a PCA plot using all parameters that were predictive of either regimen or protection status (*Figure 7B*). The information-related parameters that were predictive of protection in this model is shown in *Supplementary file 1*. Although there was some overlap between DFD responders and STD responders, the weak responders clustered close together, suggesting that they are markedly different from subjects that show promising early-stage immune responses. Finally, the difference in efficacy between the DFD and STD regimens seemed to be accounted for entirely by a subset of early strong responders that failed to achieve protection in the STD regimen.

## Discussion

Malaria is a leading cause of morbidity and mortality in endemic areas, underscoring the need for an effective vaccine. The RTS,S/AS01 vaccine is a promising candidate that has undergone extensive testing to define an optimal dosing and vaccine delivery strategy. In Malaria-071, a CHMI trial, participants in the DFD group who received a delayed and reduced third dose achieved 86% efficacy, which was significantly greater than the 62.5% protection attained under the STD regimen in which three vaccine doses are given at monthly intervals (*Regules et al., 2016*). To understand whether there was an immunologic basis to explain this difference, we conducted a study to examine T–B cell interactions in PBMC obtained from timed blood samples in the two regimens. For T cells, our focus was on delineating the dynamics of CSP antigen-specific pTfh cells, which were defined by phenotype and function. For B cells, we examined B cell maturation markers to define subsets and evaluated their function. A data integration approach was used to define correlates of vaccine-induced

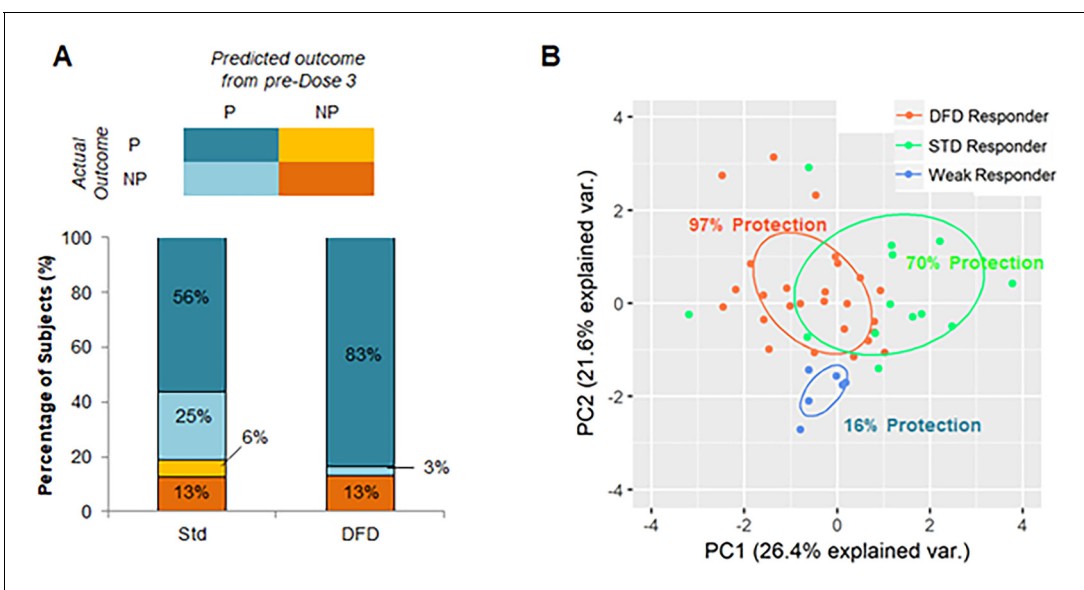

**Figure 7.** Identification of subjects with promising early-stage immune responses. (**A**) Comparison of STD and DFD subjects in terms of their predicted outcome from early-stage (pre-dose 3) immune response data and their actual outcomes. (**B**) PCA using immune parameters that are predictive of regimen and protection-status as determined by machine learning. Subjects are color-coded on the basis of their classification (on the basis of early-stage [pre-dose 3] immune data) as DFD responders (DFD subjects predicted to be protected), STD responders (STD subjects predicted to be protected), and weak responders (subjects predicted to be not protected). Protection rate is shown as the percentage of subjects in each group that was found to be protected in the study.

protection and non-protection. We found that protected subjects in both vaccine regimens were characterized by early induction of CSP antigen-specific pTfh responses, followed by functional memory B cell responses preceding the third dose that persisted at later time points. The non-protected subjects in both the DFD and STD regimens failed to mount the early pTfh response or B cell responses, pointing to the importance of pTfh in vaccine-induced protection. A key finding that provided insight into the inferiority of STD regimen was that in some NP subjects an initial 'protective' type of immune response was elicited by doses 1 and 2, but was aborted following the third vaccination dose. Understanding the mechanisms by which delaying and reducing the third antigen dose of RTS,S/AS01 after initial priming/boost helps to preserve the B cell immunity will lead to improved vaccination strategies. Our study was performed in a controlled setting with uninfected adult volunteers. Only field trials can tell us how such strategies will translate in endemic areas of the world where malaria exposure is rampant.

We noticed a clear increase in total pTfh cells and CSP-specific pTfh cells as early as 6 days post-first dose of RTS,S/AS01 vaccination that occurred only in participants who were protected following experimental challenge with *P. falciparum*. Circulating pTfh cells provide a snapshot of Tfh at the lymphoid inductive sites (*Vella et al., 2019*; *Pahwa, 2019*). Studies in healthy adults have documented the importance of pTfh expansion at day 7 or day 28 post-influenza and after other vaccines as well as bouts of various infectious diseases (*Bentebibel et al., 2016*; *Bentebibel et al., 2013*; *Herati et al., 2017*; *Herati et al., 2014*; *Pallikkuth et al., 2017*; *Pallikkuth et al., 2012*; *Pallikkuth et al., 2019*; *Boswell et al., 2014*; *Cubas et al., 2013*; *Locci et al., 2013*; *Simpson et al., 2010*; *Ueno, 2016*). The pTfh expansion was noted in both the DFD and STD regimens and was sustained throughout the vaccination schedule. This observation is reminiscent of the early induction of Ebola virus-specific pTfh following a single dose of the rVSV-Zaire Ebolavirus (ZEBOV) vaccine in an endemic population in Guinea, which was associated with protection (*Farooq et al., 2016*).

We identified pTfh phenotypically by CXCR5 expression in memory CD4 T cells. Cellular markers that have been used to define pTfh cells and their subsets have varied, with CXCR5 being a universally accepted receptor on these cells (*Vella et al., 2019*; *Pahwa, 2019*; *Bentebibel et al., 2016*; *Herati et al., 2017*; *Locci et al., 2013*; *Pallikkuth et al., 2012*; *Heit et al., 2017*; *He et al., 2013*; *Morita et al., 2011*; *Moysi et al., 2018*). Tfh cells that are located in the GC of secondary lymphoid organs (*Vella et al., 2019*; *Pahwa, 2019*; *Herati et al., 2017*; *Herati et al., 2014*; *Heit et al., 2017*) express high levels of PD-1. The frequency of PD-1$^+$ cells is very low in circulating pTfh cells and the use of PD-1 as an essential marker that defines pTfh is likely to limit the frequency of identifiable pTfh. Moreover, the relevance of PD-1-expressing pTfh remains unclear because the molecule was found to be inhibitory for pTfh function in some studies (*de Armas et al., 2017a*). We have found that the expression of PD-1 in combination with that of the activation markers CD38 and HLADR in pTfh is inhibitory for their function in healthy volunteers given influenza vaccine (*Pallikkuth et al., 2019*). In light of this information, we opted to focus on the antigen-specificity of pTfh cells to mark functional cells. An important aspect of our analysis was the determination of CD40L expression, intracellular IL-21 production and ICOS upregulation in antigen-stimulated pTfh. Using these criteria, we observed higher CSP-specific pTfh in P subjects throughout the study, indicating a critical role of functional pTfh cells for protection against *Plasmodium* infection. IL-21 is the signature cytokine of Tfh cells and is required for optimal B cell function (*Crotty, 2011*; *Linterman and Vinuesa, 2010*; *Vogelzang et al., 2008*; *Bryant et al., 2007*; *Pallikkuth and Pahwa, 2013*). Likewise, ICOS–ICOSL interactions are known to be important for Tfh–B cell collaboration and also for *IL-21* gene transcription in Tfh cells through *c-Maf* (*Bossaller et al., 2006*; *Bauquet et al., 2009*). These results add to the growing body of evidence pointing to the importance of IL-21 (*Schultz et al., 2016*; *Spensieri et al., 2016*; *de Armas et al., 2017b*) and ICOS (*Bentebibel et al., 2016*; *Bentebibel et al., 2013*; *Herati et al., 2014*; *Havenar-Daughton et al., 2016*) in circulating CD4 T cells or CSP-specific pTfh as biomarkers of vaccine responses. In the data integration analysis, neither the non-pTfh cells nor IFNγ+ CSP-specific CD4 and pTfh responses were identified as variables that were associated with protection or regimen difference, reaffirming the central role of antigen-specific pTfh cells in the CD4 T cell compartment in the response to vaccination. The actual timing, magnitude and duration of the pTfh response that needs to be elicited in order to generate qualitatively superior B cell responses for improved protection in the DFD regimen will require further investigations.

To understand the nature of the B cell response to the vaccine, we examined changes in specific subsets in the context of protection and regimen. The development of strong, functionally competent CSP-specific memory B cells after the second dose in protected subjects suggests the development of highly Ag-experienced functional memory B cells following the early pTfh response induced by vaccination. Development of a CSP-specific memory B cell compartment is in line with previous studies on antibody affinity, B cell somatic hypermutation, and antibody function, and suggests that affinity maturation is altered to some extent in the DFD regimen (*Regules et al., 2016*; *Chaudhury et al., 2017*). The stronger vaccine-induced memory B cell responses elicited towards PF-16 (the C-terminus of the CSP protein) over R32LR (the central repeat region) may have played a role in protection, as this region is implicated in the initial entry of the sporozoites into hepatocytes (*Ramasamy, 1998*). In our study, we found persistence of the vaccine-induced CSP-specific memory B cells up to 159 days post-challenge, the last time point for analysis. In a previous study, Pepper and colleagues reported persistence of *Plasmodium*-specific memory B cell populations that had been induced by protein immunization up to 340 days post-infection (*Krishnamurty et al., 2016*). Longer follow-up studies are needed to understand whether the vaccine-induced persistence of pTfh responses favors this B cell response in the DFD regimen. Interestingly, aMBC were also increased in the DFD regimen, compared to the STD regimen, at T5 and T6. These cells, originally described as an exhausted subset of memory B cells in HIV infection (*Moir and Fauci, 2009*; *Moir et al., 2008*), have been found in the circulation of *Plasmodium*-infected individuals from endemic countries (*Ly and Hansen, 2019*). A role for aMBC in malaria immunity has been suggested on the basis of the accumulation of this sub-population in situations of parasitemia or shorter exposure history (*Weiss et al., 2009*; *Changrob et al., 2018*) in children and adults from malaria endemic areas (*Weiss et al., 2009*; *Changrob et al., 2018*; *Portugal et al., 2015*); importantly, aMBC were maintained in situations of persistent parasite exposure (*Ayieko et al., 2013*) with a decline over 12 months in the absence of transmission. The functional significance of vaccine-induced aMBC expansion and the role of these cells in the development of immunity to malaria needs further investigation.

A major objective of the present study was to determine the mechanism behind the improved protection in the DFD vaccination regimen as compared to the STD regimen. As both the STD and DFD groups received the same regimen during the first two doses, differences between the study arms were expected only after the second dose, when the regimens split into either a reduced third dose at 7 months in the DFD group or a full third dose one month after the second dose in the STD group. To explore how the DFD regimen enhances efficacy, we used machine-learning tools and made individualized predictions of protection on the basis of early immune responses (pre-dose 3) alone. Using this analysis, we were able to identify a group of non-protected participants in the STD regimen who showed a promising immune response after doses 1 and 2, but lost this response after the third dose. These data suggest that dose 3 in the STD dose schedule has an adverse effect on an otherwise promising immune response generated by the first two doses. This effect could be caused by the Ag or adjuvant concentration of the full third dose, or by the one-month spacing between dose 2 and dose 3, which may hinder the selection of high-affinity B and T cell clones, either through overstimulation and anergy, or through weak selection pressure resulting from high Ag availability (*Alexander-Miller et al., 1996*).

Our data on ex vivo frequencies of dead cells did not differ significantly between the DFD and STD regimens immediately after (day 6) of the third dose (data not shown), and refute the possibility of a higher rate of cell death in the STD regimen. Our results suggest that NP subjects in the STD regimen are a mix of two classes of subjects, true weak responders and strong responders that aren't protected. This mixed population may explain why it has been difficult to find correlates of protection in RTS,S studies (*Ockenhouse et al., 2015*). By contrast, in the new DFD regimen, NP subjects are almost entirely of one class — weak responders. On the basis of the overall findings, we suggest that the early Tfh response, induced by the initial two vaccine doses, results in the formation of a strong high-affinity memory B cell pool that is specific to CSP antigen; in the DFD, this leads to the expansion and differentiation of the pre-formed memory B cells to Ab-secreting cells. By contrast, in the STD vaccine regimen, early administration of the booster dose was detrimental to the expansion and differentiation of pre-formed memory B cells. Previous reports found an association of CSP-specific IL-2$^+$, TNF or IFNg$^+$ CD4 T cells or Th1 responses with vaccine responses (*Kester et al., 2009*; *Lumsden et al., 2011*). These studies did not examine CSP-specific pTfh within

the memory CD4 T cell compartment, and most probably included both pTfh and non-pTfh cells in their analysis. Our data show that within the Ag-specific CD4 T cell compartment, pTfh cells but not non-pTfh show a kinetic and functional response to the malaria vaccine.

It should be noted that our study had limitations. Only a small number of participants (4/30) became infected in the DFD regimen, precluding comparisons of infected and protected subjects within each study regimen. We also did not examine CSP-specific pTfh for their Th1 versus Th2 phenotype or for response to individual CSP peptides in order to fine-map the pTfh responses. In a recent influenza vaccine study, we found that vaccine non-responders were polarized towards an inflammatory Th1/Th17 phenotype with predominant production of inflammatory cytokine TNF and Tfh antagonistic cytokine IL-2, while in responders, pTfh cells showed a Th2 phenotype with ICOS upregulation and IL-21 production (*Pallikkuth et al., 2019*). Another study has documented a negative impact of CXCR3+, a marker of Th1 type pTfh, on antibody quantity and quality in a vaccine trial involving RTS,S/AS01B (*Bowyer et al., 2018*). More detailed characterization of the functional and phenotypic heterogeneity of pTfh in future malaria vaccine studies may be informative. Further analyses are needed to ascertain the relationships of the immune parameters investigated herein with the magnitude and breadth of the Ab responses.

We conclude that delaying and reducing the third vaccine dose is advantageous for developing a protective immune response. This is highlighted particularly in those individuals that elicited promising responses after the first two doses, which then seemed to be disrupted when the third dose of RTS,S/AS01 was administered at the standard concentration one month after the second dose. We recognize that the CHMI studies of RTS,S-vaccinated malaria-naive adults represent a controlled setting for the study of immune response in relation to vaccine-induced protection. Whether the DFD regimen can be translated into the field with beneficial effects remains to be seen. The long interval between the second and third doses may be challenging in the face of overwhelming exposure to mosquitoes that can inoculate sporozoites into the host in this period, thereby affecting the development of immunity. However, the amount of CSP delivered via natural infection is much lower than the amount of CSP in RTS,S/AS01 and, thus, it is unlikely that natural infection would disrupt the development of an otherwise protective immune response. Recently, it was shown that children who responded well to RTS,S/AS01 vaccination had increased baseline frequencies of antibody-secreting and Tfh cells (*Hill et al., 2020*). Nevertheless, emerging data also suggests that malaria infection may induce memory Tfh cells that have impaired B cell helper function, and may inhibit differentiation to fully functional Tfh cells, thus resulting in germinal center dysfunction and suboptimal antibody responses (*Hansen et al., 2017*). Last, our data indicate the generation of strong CSP-specific pTfh responses that persist even after 159 days post-challenge, suggesting that CSP-specific pTfh could serve as potential biomarkers for vaccine efficacy. Monitoring CSP-specific pTfh should be considered in future malaria vaccine trials in clinical settings or field studies.

# Materials and methods

**Key resources table**

| Reagent type (species) or resource | Designation | Source or reference | Identifiers | Additional information |
|---|---|---|---|---|
| Peptide, recombinant protein | Staphylococcal enterotoxin B | List Biological laboratories | # 122 | 1 µg/ml, PBMC stimulation |
| Peptide, recombinant protein | CPG ODN 2016 | In-Vivogen | # tlrl2006-1 | 1 µg/ml, PBMC stimulation |
| Peptide, recombinant protein | HBs peptide pool | PATH's Malaria Vaccine Initiative | | 2 µg/ml, PBMC stimulation |
| Peptide, recombinant protein | CSP peptide pool | PATH's Malaria Vaccine Initiative | | 2 µg/ml, PBMC stimulation |

*Continued on next page*

*Continued*

| Reagent type (species) or resource | Designation | Source or reference | Identifiers | Additional information |
|---|---|---|---|---|
| Peptide, recombinant protein | CS repeat region protein (R32LR) | GSK | | 2 µg/ml, PBMC stimulation |
| Peptide, recombinant protein | HBs protein | GSK | | 2 µg/ml, PBMC stimulation |
| Peptide, recombinant protein | CSP protein | PATH's Malaria Vaccine Initiative | | 2 µg/ml, PBMC stimulation |
| Peptide, recombinant protein | C-terminal peptide (PF-16) | PATH's Malaria Vaccine Initiative | | 2 µg/ml, PBMC stimulation |
| Biological sample (*Homo-sapiens*) | Primary human mononuclear cells | GSK | | Cryopreserved in liquid nitrogen |
| Antibody | Anti-human CD3 BUV 395, mouse monoclonal, Clone SK7 | BD Biosciences | RRID:AB_2744382; Cat# 564001 | 5 µl/test, FACS |
| Antibody | Anti-human ICOS BV 421, regimenenian hamster monoclonal, Clone C398.4A | Biolegend | RRID:AB_2562545; Cat# 313524 | 0.156 µl/test, FACS |
| Antibody | Anti-human CXCR5 Alexa 647, rat monoclonal, Clone RF8B2 | BD Biosciences | RRID:AB_2737606; Cat# 558113 | 0.625 µl/test, FACS |
| Antibody | Anti-human CD8 Alexa 700, mouse monoclonal, Clone RPA-T8 | BD Biosciences | RRID:AB_10643765; Cat# 561453 | 2.5 µl/test, FACS |
| Antibody | Anti-human CCR7 PE-CF594, mouse monoclonal, Clone 150503 | BD Biosciences | RRID:AB_11153301; Cat# 562381 | 5 µl/test, FACS |
| Antibody | Anti-human CD28 PE-Cy5, mouse monoclonal, Clone CD28.2 | Biolegend | RRID:AB_314312 Cat# 302910 | 2.5 µl/test, FACS |
| Antibody | Anti-human CD45RO FITC, mouse monoclonal, Clone UCHL1 | Beckman Coulter | Cat# IM1247U | 8 µl/test, FACS |
| Antibody | Anti-human CD4 PerCP-Cy5.5, mouse monoclonal, Clone L200 | BD Biosciences | RRID:AB_394488 Cat# 552838 | 2.5 µl/test, FACS |
| Antibody | Anti-human CD40L BV 605, mouse monoclonal, Clone 24–31 | Biolegend | RRID:AB_2563832 Cat# 310826 | 0.625 µl/test, FACS |
| Antibody | Anti-human Ki-67 BV 711, mouse monoclonal, Clone Ki-67 | Biolegend | RRID:AB_ 2563861 Cat# 350516 | 1.25 µl/test, FACS |
| Antibody | Anti-human CD69 APC-Cy7, mouse monoclonal, Clone FN50 | Biolegend | RRID:AB_314849 Cat# 310914 | 0.156 µl/test, FACS |
| Antibody | Anti-human IL-21 PE, mouse monoclonal, Clone 12-7219-42 | eBioscience | RRID:AB_1582260 Cat# 12-7219-42 | 0.156 µl/test, FACS |
| Antibody | Anti-human IFN-g PE-Cy7, mouse monoclonal, Clone B27 | BD Biosciences | RRID:AB_396760 Cat# 557643 | 2.5 µl/test, FACS |

*Continued on next page*

*Continued*

| Reagent type (species) or resource | Designation | Source or reference | Identifiers | Additional information |
|---|---|---|---|---|
| Antibody | Anti-human IgG BV 421, mouse monoclonal, Clone G18-145 | BD Biosciences | RRID:AB_2737665 Cat# 562581 | 2.5 μl/test, FACS |
| Antibody | Anti-human CD80 BV 605, mouse monoclonal, Clone 2D10 | Biolegend | RRID:AB_11123909 Cat# 305225 | 0.625 μl/test, FACS |
| Antibody | Anti-human IL-21R APC, mouse monoclonal, Clone 2G1-K12 | Biolegend | RRID:AB_2123988 Cat# 347808 | 2.5 μl/test, FACS |
| Antibody | Anti-human CD20 Alexa 700, mouse monoclonal, Clone 2H7 | Biolegend | RRID:AB_493753 Cat# 302322 | 0.625 μl/test, FACS |
| Antibody | Anti-human CD38 PE, mouse monoclonal, Clone HIT2 | BD Biosciences | RRID:AB_395853 Cat# 555460 | 2.5 μl/test, FACS |
| Antibody | Anti-human CD21 PE-Cy5, mouse monoclonal, Clone Bly4 | BD Biosciences | RRID:AB_394028 Cat# 551064 | 3.75 μl/test, FACS |
| Antibody | Anti-human CD10 PE-Cy7, mouse monoclonal, Clone HI10A | BD Biosciences | RRID:AB_400216 Cat# 341092 | 2.5 μl/test, FACS |
| Antibody | Anti-human IgD FITC, mouse monoclonal, Clone IA6-2 | Biolegend | RRID:AB_10612567 Cat# 348206 | 0.625 μl/test, FACS |
| Antibody | Anti-human CD27 BV 650, mouse monoclonal, Clone L128 | BD Biosciences | RRID:AB_2744352 Cat# 563228 | 2.5 μl/test, FACS |
| Antibody | Anti-human ICOSL biotin, mouse monoclonal, Clone 2D3 | Biolegend | RRID:AB_528729 Cat# 309406 | 1.25 μl/test, FACS |
| Antibody | Anti-human streptavidin BV 711, mouse monoclonal | Biolegend | Cat# 405241 | 0.7 μl/test, FACS |
| Antibody | Anti-human Ki-67 PerCP-Cy5.5, mouse monoclonal, Clone B56 | BD Biosciences | RRID:AB_10611574 Cat# 561284 | 2.5 μl/test, FACS |
| Antibody | Anti-human CD27 PerCP-Cy5.5, mouse monoclonal, Clone MT271 | Bioegend | RRID:AB_2561906 Cat# 356408 | 0.312 μl/test, FACS |
| Antibody | Anti-human CD28 unconjugated, mouse monoclonal, Clone L293 | BD Biosciences | RRID:AB_400197 Cat# 340975 | 1 μg/ml, PBMC stimulation |
| Commercial assay or kit | Human IgG ELISA Quantitation Set | Bethyl Laboratories | Cat# E80-104 | IgG ELISA, PBMC culture supernatants |
| Chemical compound, drug | Brefeldin A | Sigma Aldrich | Cat# B7651-5mg | 10 μg/ml, PBMC stimulation |
| Software, algorithm | FlowJo | BD Biosciences | | https://www.flowjo.com |
| Software, algorithm | Prism 8 | GraphPad | | https://www.graphpad.com/scientific-software/prism/ |
| Software, algorithm | BD FACSDiva | BD Biosciences | | https://www.bdbiosciences.com/en-us/instruments/research-instruments/research-software/flow-cytometry-acquisition/facsdiva-software |

*Continued on next page*

*Continued*

| Reagent type (species) or resource | Designation | Source or reference | Identifiers | Additional information |
|---|---|---|---|---|
| Other | LIVE/DEAD Fixable Aqua Dead Cell Stain | Invitrogen | L34957 | 0.5 µl/ml, FACS |

## Study timepoints and processing

Recruitment of participants, vaccine administration and CHMI studies were conducted at Walter Reed Army Institute of Research (WRAIR) (*Regules et al., 2016*). A schema for vaccine timepoints and blood sample collection for immunology analyses in both vaccine regimens and in the control regimen of the study is shown in *Figure 1*. Blood draws for this were performed at eight different timepoints designated T0–T7: (i) pre-vaccination (T0), day 6 post dose 1 (T1), day 28 post dose 1 (T2), day 6 post dose 2 (T3), day 28 post dose 2 (T4), day 6 post dose 3 (T5), day 21 post dose 3 and pre-challenge (T6) and at study end (T7), 159 days post-challenge. Blood was processed for peripheral blood mononuclear cells (PBMC) and cryopreserved PBMC samples were shipped to the University of Miami (UM) in liquid nitrogen. All lab testing, analysis, data entry and plotting of graphs were performed in batches in a blinded manner with information related to vaccine regimen and protection status revealed to the UM lab only after all of the samples had been processed. This study was approved by the Institutional Review Boards of UM. De-identified PBMC samples were processed at UM laboratory.

## Vaccine antigens, control antigen and monoclonal Abs

The HBs peptide pool, CSP peptide pool, CS repeat region protein (R32LR) and HBs protein were provided by GSK. The CSP protein (PF-CSP) and the C-terminal peptide (PF-16) were provided by PATH's Malaria Vaccine Initiative (MVI). Staphylococcal enterotoxin B (List Biological Laboratories) and CPG ODN 2006 (InvivoGen) were used as positive control antigens for T and B cells, respectively. A list of monoclonal Abs used for the flow cytometry studies, including the information about clone and fluorochrome, is included in the Key resources table.

## pTfh frequency and antigen-specific intracellular cytokine secretion (ICS) in short-term cultures

pTfh frequency and function were analyzed at pre-vaccination (T0) and at all timepoints post-vaccination in each regimen. Briefly, PBMC ($1.5 \times 10^6$/ml/condition) were stimulated with CSP peptide pool (2 µg/ml) and HBs peptide pool (2 µg/ml) along with co-stimulation molecule antiCD28 (1 µg/ml) for 12 hr at 37°C. Brefeldin A (10 µg/ml) was added 7 hr after stimulation to prevent protein transport. SEB (1 µg/ml) was used as a positive control, and medium with co-sitm molecule antiCD28 alone was used as a negative control. After stimulation, cells were stained using a 14-color fluorochrome conjugated monoclonal Ab panel including surface markers that are specific for pTfh identification along with live/dead amine dye (aqua). Cells were then fixed, permeabilized and stained intracellularly for interleukin (IL-21), CD40L, and Ki67, acquired on a BD LSRFortessa, and analyzed by FlowJo (v 9.4.3, Tree Star Inc).

All flow cytometry analyses were performed using optimally titrated Ab concentrations and after applying appropriate fluorescent compensation using DIVA software on BD LSRFortessa at the time of acquisition and fine tuning of compensation using FlowJo at the time of analysis. Gating controls included unstained cells and Fluorescence Minus One (FMO), and for all stimulation experiments, unstimulated cells were used as additional biological controls. For basic pTfh identification, CD4 T central memory ($T_{CM}$[CD3$^+$CD4$^+$CD45RO$^+$CD27$^+$]) were gated on CXCR5 to determine the frequencies of CXCR5$^+$ subsets in CD4 $T_{CM}$ cells, designated as pTfh cells. For Ag-specific pTfh cells, CD40L$^+$ CD4 T cells were gated on the basis of the expression of CD45RO and CXCR5 (as CD45RO$^+$CXCR5$^+$) as CSP-specific pTfh cells and CD45RO$^+$CXCR5$^-$ as CSP-specific non-pTfh cells (*Figure 2—figure supplement 1*). CSP-specific pTfh cells and non-pTfh cells were further analyzed for the intracellular expression of signature cytokine IL-21, inducible co-stimulator ICOS and proliferation marker Ki67.

### Ex vivo B cell maturation subsets

Thawed PBMC were analyzed for B cell phenotypes without in vitro stimulation by flow cytometry. Total mature B cell were identified as CD3$^-$CD10$^-$CD20$^+$ cells after excluding immature CD10$^+$ B cells, and total memory B cells were identified as CD20$^+$CD27$^+$ cells. On the basis of the expression of CD21, CD27 and IgD, B cell maturation subsets were identified as naïve $^{(CD21hiIgD+CD27-),}$ resting memory (RM: CD21$^{hi}$CD27$^+$), activated memory (AM: CD21$^{low}$CD27$^+$) and atypical memory B cells (aMBC: CD21$^{low}$CD27$^-$). Within the total memory, RM and AM B cells IgD$^+$IgG$^-$ were identified as unswitch and IgD$^-$IgG$^+$ as switch memory B cells (*Figure 5—figure supplement 1*).

### Spontaneous Ab-secreting cells (ASC)

A spontaneous ASC enzyme linked immunospot (ELISpot) assay was performed at T0, T1, T3, and T5 as described previously (*George et al., 2015*; *Rinaldi et al., 2017*; *Pallikkuth et al., 2011b*) against wells coated with PF-CSP, R32LR and PF-16 malaria antigens, as well as HBs-antigen, using unstimulated PBMC. Data are expressed as spontaneous ASC/million PBMC.

### Memory B cell analysis

Memory B cell responses were analyzed at T0, T2, T4, T5, T6 and T7 using memory B cell antibody secreting cell (ASC) ELISpot (*George et al., 2015*; *Rinaldi et al., 2017*; *Pallikkuth et al., 2011b*). PBMC 1.5 × 10$^6$/ mL per condition were stimulated for 5 days with 2 µg/ml each of malaria (PF-CSP, PF-16, R32LR) and HBs antigens. CpG oligodeoxynucleotides 2006 (CpG ODN2006: 1 µg/ml) was used as a positive control and medium as a negative control. On day 5, cells were harvested and assayed for Ag-specific induction of IgG-secreting cells by ELISpot assay (*Pallikkuth et al., 2011b*). The remaining cells were stained with fluorochrome conjugated monoclonal Abs that were specific for B cell maturation subsets (naïve, total memory, RM, AM, and aMBC). Switch and unswitch memory B cells within total memory, RM and AM subsets were identified on the basis of IgG and IgD expression along with proliferation marker Ki67 and analyzed by flow cytometry as described in *Figure 5—figure supplement 1*. Culture supernatants were stored at −80°C and assayed for IgG by ELISA.

### Statistical analysis and data integration

To analyze this complex immunology dataset, which includes a large number of immune measurements for each subject, we performed an integration approach in which we combined traditional univariate analysis with multivariate machine-learning methods to isolate immune responses that were vaccine induced, to characterize regimen-specific differences, and to identify correlates of protection. A general workflow of the data integration approach is shown in *Figure 6—figure supplement 2*. Analyses were performed to compare changes in pTfh and B cell related markers for each group between different timepoints pre- and post-vaccination, or between P vs NP, or between regimens DFD and STD, at each timepoint or at selected timepoints. Generalized linear mixed-effects models (GLMM), fitted via Penalized Quasi-Likelihood (PQL) using R 'MASS' package was used to accommodate repeated measures of time, with random intercept set by patient ID (PID). P value was adjusted for multiple comparisons by Benjamini and Hochberg correction using R 'multcomp' package. A p value of <0.05 was considered to be significant. Immune measures were classified as vaccine-induced responses if they showed a significant difference from the pre-immune (*vs*. T0) timepoint. Immune measures were classified as regimen-specific and protection-specific differences if they showed a significant difference with respect to vaccine regimen (STD vs. DFD) or protection status (P vs. NP), respectively.

To assess the predictive value of the regimen- and protection-specific differences identified in this study, we used the random forest model, a machine-learning method, to make individualized predictions of regimen or protection status on the basis of the immune data alone. The random forest model was generated using all vaccine-induced immune responses (R *caret* package). We trained the model using the repeated cv method, subsampling the dataset by five-fold and resampling ten times. The random forest model was tuned using the caret R package. Specifically, the number of branches of the tree (*mtry*) and the rule for splitting (*gini* or *extratrees*) were adjusted to identify the optimal accuracy and kappa values during internal ten-fold cross validation, repeated ten times. The *oneSE* method was used to select the optimal model. To test the predictive accuracy of the random

forest modeling approach, we carried out a leave-one-out analysis, in which one subject was removed from the dataset, after which the model was trained on the remaining subjects and then used to predict the adjuvant condition of the excluded subject on the basis of its immune data. We performed this for all subjects in the dataset, and calculated both the accuracy and kappa value of the prediction model. We used the *varImp* function to determine the variable importance for each generated model, and reported the average variable importance across all models to assess the relative importance of each vaccine-induced immune measure to predicting regimen or protection status.

Principal component analysis (PCA) was carried out in R using the *ir.pca* package and visualized using *ggbiplot*. For the PCA, we used a subset of the immune measures, using only parameters that were found to be predictive of regimen or protection status, as determined by variable importance analysis in the random forest model. Finally, correlation analysis was carried out in R using the *cor* function to calculate the Pearson correlation coefficient. All immune measures were compared with all other immune measures, and the correlation matrix was used for hierarchical clustering, using the *hclust* function in R, to identify groups of correlated immune parameters, termed 'immune clusters'. All immune parameters within an immune cluster have a Pearson correlation coefficient of at least 0.80 to every other parameter in that cluster.

## Acknowledgements

We thank Maria Pallin, Celeste Sanchez and Varghese George from the Pahwa lab for their assistance with laboratory experiments, Dr Ann-Marie Cruz for project management support and scientific input, GSK for PBMC samples, and the study participants. This work was supported by PATH's Malaria Vaccine Initiative or PATH, the US Army Military Infectious Disease Research Program, and the U.S. Army Medical Research and Development Command. The opinions and assertions contained herein are the private views of the authors and are not to be construed as official or as reflecting the views of the US Army, the US Department of Defense, or The Henry M Jackson Foundation for the Advancement of Military Medicine, Inc. This paper has been approved for public release with unlimited distribution.

## Additional information

### Competing interests

Erik Jongert: is an employee of GSK, and owns shares of GSK. The other authors declare that no competing interests exist.

### Funding

| Funder | Author |
| --- | --- |
| PATH's Malaria Vaccine Initiative or PATH | Savita Pahwa |

The funder contributed to conceptualization, data review and manuscript preparation

### Author contributions

Suresh Pallikkuth, Conceptualization, Data curation, Formal analysis, Supervision, Investigation, Methodology, Writing - original draft, Writing - review and editing; Sidhartha Chaudhury, Data curation, Software, Methodology, Writing - original draft; Pinyi Lu, Li Pan, Software, Formal analysis; Erik Jongert, Conceptualization, Methodology, Writing - review and editing; Ulrike Wille-Reece, Conceptualization, Resources, Supervision, Methodology, Project administration, Writing - review and editing; Savita Pahwa, Conceptualization, Supervision, Funding acquisition, Methodology, lab facilities, technical support, writing, review and editing

Author ORCIDs

Savita Pahwa (iD) https://orcid.org/0000-0002-4470-4216

Decision letter and Author response

Decision letter https://doi.org/10.7554/eLife.51889.sa1
Author response https://doi.org/10.7554/eLife.51889.sa2

## Additional files

### Supplementary files

• Supplementary file 1. Supplementary file 1A. Summary of vaccine-induced immune measures. Abbreviations: Spontaneous antibody secreting cell ELSIPOT (AELI), CSP- and HBs-specific B cell subsets by flow cytometry (BCF), frequencies and function of total pTfh, CSP-, HBs- and SEB-specific CD4 and pTfh data (Tfh ICC), CSP- and HBs-specific memory B cell ELISpot data (BELI), CSP- and HBs-specific PBMC culture supernatant IGG (IgG). Supplementary file 1B. Parameters most predictive of protection using the early-response (pre-Dose 3) immune data.

• Transparent reporting form

### Data availability

All data generated or analysed during this study are included in the manuscript and supporting files. Source data files have been provided for Figures 2, 3, 4 and 5.

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
