## [Decision Letter]

**Acceptance summary:**

RTS,S is a leading *Plasmodium falciparum* recombinant circumsporozoite protein (CSP) vaccine. Administered according to the standard 1-2-3 month regimen, RTS,S induces but a modicum of protracted protective immunity. By contrast, vaccination according to a delayed and fractional 3^rd^ dose results in a superior protective outcome in controlled human malaria infection challenge studies. Understanding vaccine-induced immune protective mechanisms is crucial for developing of an improved vaccine against malaria. In this study, the authors conducted comparative analyses of RTS,S elicited immune responses in subjects vaccinated according to the standard versus delayed and fractional doses. Analyses of peripheral blood cellular and antibody responses combined with machine relearning approaches revealed that early induction of CSP-specific pTfh CD4 T cells and the persistence of CSP-specific B cell responses are associated with the superior protective outcome in the group that received delayed and fractional dose of RTS,S/AS01.

**Decision letter after peer review:**

Thank you for sending your article entitled "Cellular biomarkers of antibody response to RTS,S/AS01 malaria vaccine in a controlled human vaccine challenge model" for peer review at *eLife*. Your article is being evaluated by three peer reviewers, one of whom is a member of our Board of Reviewing Editors, and the evaluation is being overseen Dominique Soldati-Favre as the Senior Editor.

The reviewers considered that the authors indeed posed and important question regarding the mechanism of RTS,S-mediated protection in a very systematic fashion. On the basis of the reviews provided below, the main concerns are as follows: issues with flow cytometry results for both T and B cells; the accompanying statistical analysis; inclusion of fine/distinct CSP protein peptide or region specificities of IgG and pTfh cells; correlative interaction between pTfh and B cells from individuals, which would address the relationship and/or the qualitative difference between these cell types in protected DFD vs. STD; the machine learning portion needs to be properly presented/explained in the result section; revisions in the labeling of certain figures for clarity; and, finally, please address the issue concerning the writing style, particularly in the Results section.

Reviewer #1:

RTS, S is the most advanced vaccine against pre-erythrocytic (sporozoite) stage of malaria. Studies conducted in malaria endemic areas show only a modicum of protective efficacy and short durability. On the basis of the results from an earlier study by Stoute et al., the repeat of the approach of delivering RTS,S as DFD vs. STD immunizing dose has shown much promise in the recently conducted CHMI studies. The current study describes response of peripheral CD4 T cells, pTfh cells, and B cells as correlates of protection. The results add much value as they expand our understanding of the immune responses that form the basis of protection against malaria. Although immunological parameters may be quite different in endemic areas, particularly among older children who have had malaria infection prior to RTS,S vaccination, nonetheless, this study provides new information regarding pTfh cells in protective immunity against CHMI.

As with any studies detailing immune responses in human PBMC samples, this study was admittedly quite challenging; the authors succeeded in performed carefully timed analyses and amassed lots of important results. Applying machine learning analyses, they have deduced predictive value concerning early pTfh cells and memory B cells responses as predictors of protection. The manuscript is well organized and, on the whole, the experiments are well performed; however, there are several critical issues that need to be addressed.

The narrative in the Results section provides data without any benefit of a proper introduction as to what specific question was being asked and why a particular analysis was undertaken. For example, why was ICOS evaluated? Why were B cell responses evaluated? Similarly, what is the significance of measuring cellular expression of Ki67? The exception was an explanation for determining IL-21 for pThf cells.

The entire Results section concerning B cells is very difficult to follow. The main reason of this is that none of the discussed B cell subsets, e.g. MB, are defined phenotypically. I do acknowledge that the phenotypic markers are indicated in the Materials and methods section; however, it is rather clumsy to go back and forth between the pages looking for the specific phenotypes in one section of the paper and then going back to the Results section for a response of a given B cell population. Even for a reader with some background in immunology, it would be difficult to understand the significance of "…vaccine induced alteration in B cell phenotypes and function…", as neither specific phenotypic markers nor functions are indicated. What exactly are recall antigens, R32LR and PF-16?

Speaking of cellular phenotypes, I have not found any gating strategy for B cells. Therefore, it is impossible to evaluate responses shown in Figures 3 and 4. The authors discuss Ig switching, yet apart from ELISPOS for IgG-producing B cells, no other isotypes are shown. It is not clear what criteria the authors have used for the discussion of B cell maturation. Starting with data in Figure 3, how were the B cell gated for the different subsets? The B cell subsets need to be clearly indicated or identified by a set of specific phenotypic markers and Ig isotypes, otherwise, the abbreviations such as sAM, SM, are meaningless for the reader.

The authors mention that Ab ELISA experiments were performed, but the authors do not show any results. Also, the markers for atypical B cells depart from what is currently described in the literature. And what was the rationale for even considering atypical memory B cells?

Another critical issue is the authors' generalization of CSP specificities of CD4 T cells. The authors did use PF CSP peptide pools for activation of CD4 or pTfh cells. The content of these pools is not shown, and it needs to be indicated. Importantly, some peptide pools representing certain regions of PF CSP might have shown more recall activity than other peptide pools. In a previously published study, Schwenk and colleagues (2011, Vaccine 29), interrogated RTS,S induced CD4 T cells responses with PF CSP peptides and they showed that subjects respond to a discrete set of PF CSP peptides. Importantly, once PF CSP peptide-specific early responses are induced, the specificity repertoire does not shift following boost immunization or CHMI. I will add, that the authors looked at inflammatory CD4 T cells. In view of these published results concerning PF CSP peptide specific inflammatory CD4 T cells responses, such information would be quite important if only to understand what specificities induced the early pTfh cells in the current study; is there specificity overlap between CD4 T cells with different functional attributes? Because according to the current observation early CD4 T cell responses do matter; it is imperative to show the results from the individual pool activated T cells. In this regard, it also would have been instructive to determine HLA of P vs. NP or early responders.

It is not entirely clear as to why the authors are surprised that the observed correlates of protection were directed to Ag-specific CD4 and pTfh cells that arose post does 1. To gain some understanding of the T-B cell collaboration for Ab production, it would have been instructive to investigate if the B cell responses in protected persons were T cell dependent. There are markers for such B cell responses, e.g., CD73, CD80 etc. These analyses might need to be conducted to sort our T cell dependent vs. T cell independent B cell responses in the P vs. NP subjects. Pepper et al. did show that *Plasmodium* induced B cells survive only for a short time as short-lived memory B cells. This may provide some explanation to the authors' query.

The labeling of the figures should be improved, particularly with regard to the time of PBMC sampling versus vaccinations. The authors need to indicate by arrows on the X axis the timing of vaccination vis-a-vis samplings. This will frame the scheduling indicated by T0, T1, etc. Otherwise, one needs to constantly consult the text for clarification.

The writing in general is a bit confusing. For example, (1) In the Discussion – "A major objective of.……was to determine if one regimen, DFD or STD, was superior…" The results of the superiority of DFD over STD in protection was published previously. Please reword the sentence; (2) "…that differences in immunity in the DFD regimen arose…" (3) "…B cell maturation subsets…"; (4) previously conducted studies are not referenced.

In the Abstract the authors state that B cells are rescued by the DFD immunization regimen. It is true that B cells producing IgG responses declined (Figure 3). Are B cell numbers also lost? I do not recall any studies where the authors show apoptosis, or some other form of B cell death induced by the STD immunization regimen.

Reviewer #2:

In this study Pallikkuth et al. examine peripheral T follicular helper (pTfh) cells and memory B cells as potential immunologic mechanisms responsible for the increased protection observed in subjects receiving RTS,S/AS01 in a delayed fractional dose (DFD) compared to the standard dose regimen (STD).

The authors found that subjects protected from malaria demonstrate an expansion of total- and CSP specific pTfh cells following vaccination and that the variation observed in protection between the STD and DFD arms arises mainly from variations in antigen-specific B cells. In addition, the authors apply a machine learning analysis to identify immune correlates associated with protection from malaria, which indicates an important role for early induction of pTfh responses and suggests that class-switched memory B cells can be engendered "saved" by a delayed fractional dose regimen. This study addresses an important question regarding the mechanism of RTS,S-mediated protection in a very systematic fashion, leveraging a unique trial dataset. The reported association with Tfh responses and protection from RTS,S is a significant advance, and the purported differences observed between the dosing regimens of high potential significance. However, my major concerns revolve around the quality of the flow cytometry data, which is difficult to assess given limited info on gating and absence of B cell flow plots. In addition, some aspects of the statistical analyses are not clear. An inherent weakness of the machine learning analyses rests in the fact that it does not use separate training, validation, and test sets. The dataset from which the model is derived appears to be the same dataset from which the authors go back and make predictions, which limits external inference. While the Introduction and Discussion are well written but several aspects of the Results and figures are not clear.

1) It is unclear from Figure 2 how flow gates were set; ideally this could be based on FMO controls, which should be shown in the gating panel, but I could not find any mention of how gates were determined in the Materials and methods. The cutoff for ICOS, IL21, and Ki67 looks rather arbitrary, and the Ki67 stain did not appear to provide good separation based on the (very small) example plot.

2) No flow cytometry plots for B cell subsets are provided. These are critical data and should be shown (with Figure 4 or as Supplemental). This is of particular concern given that the staining for Ki67 on T cells (Figure 2C) was not very convincing, and Ki67^+^ B cells comprise the majority of the most predictive parameters in Table 1.

3) What do the error bars indicate in Figure 1-4 (SD, SEM, other)? In some cases the error bars are quite short relative to the difference between P and NP values, yet the difference is not significant; while in the puzzling case of Figure 2F (T2) the difference IS statistically significant yet the values are almost the same and the bars widely overlap.

4) The definition of Tfh is somewhat non-conventional and only partially justified (i.e. in Discussion re: PD-1).

5) The reader has to work rather hard to grasp what comparisons were statistically significant. The differences from T0 to other timepoints stand out (but are not the most interesting result). This overshadows the p-values comparing P and NP. Only the latter are mentioned in the Figure Legend. (applies to Figures 1-3).

6) The machine learning portions of the paper (rationale, findings) are nicely explained in the Discussion, but are presented in a very confusing manner in the Results. I did not understand the point of this analysis until the Discussion.

7) Figure 5 was not helpful to this reviewer and could be considered for omission; conversely Figure S1 showing the trial schematic with definition of timepoints T0-T7 was very helpful and should be considered for inclusion in the main text.

8) Was any analysis of how Tfh correlate with B cells in individuals performed? This would be of interest but appears to be missing.

9) The paper would benefit from a discussion of the implications of these findings on RTS,S vaccination in real-world endemic settings, where many children would inevitably be exposed (often repeatedly) to malaria antigens during the "suboptimal" 2-6 month window following priming (see Obeng-Adjei et al., 2015).

Reviewer #3:

The manuscript entitled "Cellular biomarkers of antibody response to RTS, S/AS01 malaria vaccine in a controlled human vaccine challenge model" by Pallikkuth et al., immune response to RTS, S/AS01 malaria vaccine and found that delayed fractional dose (DFD) regimen is superior in inducing higher class-switched Ag specific memory B cells post the third immunization compared to the (STD) standard dose regimen.

Comments as follows:

1) What is the qualitative difference between the protected versus non-protected subjects in terms of the cTfh levels or its functions?

2) The authors indicate that expansion of IL-12 and ICOS^+^ Ki67^+^ cells is indicative of vaccine-induced enhancement; however, this happens in both the arms. It did not look like this is happening only in the DFD arm. Results are written as if they are significantly different.

3) Do the authors have any hypothesis for the enhancement of B and Tfh cell response in DFD group compared to the STD regimen?

4) Did they do an antigen-specific IgG subclass response to see whether DFM has wide range of antibody responses compare to STD regimen?

5) It is also surprising to see that non-Tfh component are not looked at properly such as IFN-γ, TNF-α, and IL-2 levels. What is the association of these cells with antigen specific B cell response between P versus NP individuals?

6) In the Figure 7 it is not very clear that what predicts the protection and what is not predicting the protection.

[Editors' note: further revisions were suggested prior to acceptance, as described below.]

Thank you for resubmitting your work entitled "Cellular biomarkers of antibody response to RTS,S/AS01 malaria vaccine in a controlled human vaccine challenge model" for further consideration by *eLife*. Your revised article has been evaluated by Dominique Soldati-Favre (Senior Editor) and a Reviewing Editor.

The majority of the requested revisions are editorial in nature, with few exceptions.

1) In their response to a comment on non-pTfh cells, the authors rely on negative data (not shown) derived from the integration analysis. Despite the outcome of this analysis, it might be useful and most informative to assess the non-pTfh cells alongside the pTfh cells. In the event that the PBMCs are no longer available, the authors need to address these potentially crucial components as possible mechanisms for the superior outcome in DFD regimen of immunization. As an example, previously published works have shown an association between IL-2^+^ and IFNγ^+^ CD4 T cells and antibody responses in RTS,S vaccinated subjects. In the absence of results, including these points would improve the tenor of the Discussion.

2) Please revisit the legends and confirm whether they indicate that the error bars signify SEM.

3) Figure 3. Unlike IL-21 and ICOS, the expression of Ki67 was rather transitory and this point needs to be acknowledged.

4) A title should be more reflective of the key findings.

5) In the Abstract, there should be a clearer pivot from data that is previously published (trial results) to data presented therein.

6) Subheading “CSP-specific B cell responses emerge in protected subjects after the second dose and higher regimen specific memory B cell response in DFD arm” makes no sense grammatically.

7) In the Discussion, the authors state that "This is the first study demonstrating a protective role of CSP-specific pTfh in RTS,S/AS01 vaccination". Please tone down this statement as the results do not show a protective role of pTfh cells, they simply show an association.

8) Discussion section, final two sentence in the penultimate paragraph should could be removed, as they are unnecessary.

9) Results section requires extensive editing in that a rationale for choosing to ask a certain question or perform a particular experiment needs to be clearly stated. It is insufficient to state, "We next investigated…".

10) Discussion paragraph six is very repetitive to early wording in the Results.

11) The Discussion section on the whole still appears largely as a rehash of the Results section. It fails to provide a cogent picture or a hypothesis that would suggest possible interactions that may be occurring between the pThf cells (and possibly non-pTfh CD4 T cells) and the various B cells subsets and which bring about this enhanced protection seen in subjects receiving DFD regiment.

12) Along these lines, the authors need to relate their finding to a larger picture of RTS,S vaccination in African countries. Specifically, the paper would benefit from a discussion of the implications of these findings on RTS,S vaccination in real-world endemic settings, where many children would inevitably be exposed (often repeatedly) to malaria antigens during the "suboptimal" 2-6 month window following priming. The authors are not requested to provide field data, which they obviously cannot; however, if a DFD regimen were deployed in Africa, many children would get a natural malaria infection that might have the same negative impact as the STD dose #3 and this is worth discussing.

---

## [Author Response]

Reviewer #1:[…]The narrative in the Results section provides data without any benefit of a proper introduction as to what specific question was being asked and why a particular analysis was undertaken. For example, why was ICOS evaluated? Why were B cell responses evaluated? Similarly, what is the significance of measuring cellular expression of Ki67? The exception was an explanation for determining IL-21 for pThf cells.

We have edited the Results section to address the concerns raised and indicated the relevance of investigating pTfh, ICOS, Ki67, and B cell subsets in this study. (Subsection “CSP-specific pTfh responses are elevated in protected subjects” and “CSP-specific B cell responses emerge in protected subjects after the second dose and higher regimen specific memory B cell response in DFD arm”.)

The entire Results section concerning B cells is very difficult to follow. The main reason of this is that none of the discussed B cell subsets, e.g. MB, are defined phenotypically. I do acknowledge that the phenotypic markers are indicated in the Materials and methods section; however, it is rather clumsy to go back and forth between the pages looking for the specific phenotypes in one section of the paper and then going back to the Results section for a response of a given B cell population. Even for a reader with some background in immunology, it would be difficult to understand the significance of "…vaccine induced alteration in B cell phenotypes and function…", as neither specific phenotypic markers nor functions are indicated. What exactly are recall antigens, R32LR and PF-16?

We have included the gating strategy for the different B cell subsets in Figure 5—figure supplement 1 and have added the description of B cell subsets in the Results and Materials and methods sections of the revised manuscript. We have also revised the statement to make it more specific for measuring the phenotype of B cell subsets by flow cytometry and their function by ELIspot assays. In this prime boost vaccination strategy R32LR peptide corresponds to the repeat region of PF-CSP and the PF-16 peptide corresponds to the C-terminal region of PF-CSP. The vaccine that was administered to CSP-negative adults includes both, the R32LR and the C-term region of PF-CSP.

Speaking of cellular phenotypes, I have not found any gating strategy for B cells. Therefore, it is impossible to evaluate responses shown in Figures 3 and 4. The authors discuss Ig switching, yet apart from ELISPOS for IgG-producing B cells, no other isotypes are shown. It is not clear what criteria the authors have used for the discussion of B cell maturation. Starting with data in Figure 3, how were the B cell gated for the different subsets? The B cell subsets need to be clearly indicated or identified by a set of specific phenotypic markers and Ig isotypes, otherwise, the abbreviations such as sAM, SM, are meaningless for the reader.

Please refer to Comment 2 regarding the inclusion of a detailed gating strategy for B cells in the revised manuscript. Switched and unswitched B cells were identified based on the IgG and IgD expression on CD27^+^CD21^hi^ or CD27^+^CD21^low^ B cells with IgD^+^IgG^-^ as unswitched and IgD-IgG^+^ as switched B cells. This information is now included in the Materials and methods and Results sections of the revised manuscript.

The authors mention that Ab ELISA experiments were performed, but the authors do not show any results. Also, the markers for atypical B cells depart from what is currently described in the literature. And what was the rationale for even considering atypical memory B cells?

We analyzed the IgG levels in PBMC culture supernatants after 5 days of stimulation with the following vaccine antigens: full-length CSP (PF-CSP), C-term CSP (PF-16), and repeat region (R32LR). Responses to each of these antigens were included in the data integration analysis and none of the IgG responses appeared as variables associated with protection or regimen difference. We have now included the IgG data as a supplementary figure (Figure 5—figure supplement 5) in the revised manuscript.

In this study, we identified atypical memory B cells based on their expression of CD20, CD21, and CD27 (CD20^+^ CD21^lo/neg^ CD27^-^B cells). A similar phenotype of atypical memory B cells (aMBC) was reported previously by different investigators. A higher frequency of aMBC have been reported in children and adults with chronic asymptomatic *Plasmodium falciparum* infection, suggesting an expansion of these B cells in malaria endemic areas (Weiss et al., 2009; Portugal et al., 2015). A decline in the atypical MBC pool was observed during a 12-month period without malaria transmission, further substantiating a role of persistent parasite exposure in the maintenance of this population (Ayieko et al., 2013). In the present study, our goal was to investigate antigen-specific atypical memory B cells following RTS.AS01 vaccination and their association with vaccine-induced protection, Moreover, we wanted to assess effect of DFD vs. STD regimen on the frequencies of vaccine-specific aMBC. We found higher frequencies of PF-CSP-specific proliferating (Ki67^+^) aMBC at T5 and T6 in the DFD regimen compared to STD regimen. We have discussed this in the Discussion section.

Another critical issue is the authors' generalization of CSP specificities of CD4 T cells. The authors did use PF CSP peptide pools for activation of CD4 or pTfh cells. The content of these pools is not shown, and it needs to be indicated. Importantly, some peptide pools representing certain regions of PF CSP might have shown more recall activity than other peptide pools. In a previously published study, Schwenk and colleagues (2011, Vaccine 29), interrogated RTS,S induced CD4 T cells responses with PF CSP peptides and they showed that subjects respond to a discrete set of PF CSP peptides. Importantly, once PF CSP peptide-specific early responses are induced, the specificity repertoire does not shift following boost immunization or CHMI. I will add, that the authors looked at inflammatory CD4 T cells. In view of these published results concerning PF CSP peptide specific inflammatory CD4 T cells responses, such information would be quite important if only to understand what specificities induced the early pTfh cells in the current study; is there specificity overlap between CD4 T cells with different functional attributes? Becasue according to the current observation early CD4 T cell responses do matter; it is imperative to show the results from the individual pool activated T cells. In this regard, it also would have been instructive to determine HLA of P vs. NP or early responders.

Experiments related to fine-mapping with distinct CSP peptides will not be possible at this point and are out of scope for the current study. We achieved the goals of the present study which were to identify RTS,S/AS01vaccine-induced pTfh and B cells and their relation to malaria infection outcome in a controlled human malaria infection model. Moreover, we don’t have cryopreserved cells available for additional experiments, as blood sample collection was limited to specific objectives of the study. In the context of the comment being raised here, the review by Moris et al. (Hum Vaccin Immunother. 2018;14:17-27) discusses CSP epitope-specific CD4 T cell responses in malaria vaccine trials in prior studies performed by GSK and others. The serum IgG response from the current trial was published previously by our collaborators (Chaudhury et al., 2017).

Importantly, we included CSP antigen-induced intracellular IFNγ responses by total CD4 and pTfh cells, as well as CSP-specific pTfh cells in the data integration analyses and our results show that none of the IFNγ^+^ T cell variables were associated significantly with protection or regimen level differences. HLA information of protected and non-protected subjects or early responders were not available at the time these analyses were conducted. We mentioned this in the Discussion section as a study limitation.

It is not entirely clear as to why the authors are surprised that the observed correlates of protection were directed to Ag-specific CD4 and pTfh cells that arose post does 1. To gain some understanding of the T-B cell collaboration for Ab production, it would have been instructive to investigate if the B cell responses in protected persons were T cell dependent. There are markers for such B cell responses, e.g., CD73, CD80 etc. These analyses might need to be conducted to sort our T cell dependent vs. T cell independent B cell responses in the P vs. NP subjects. Pepper et al. did show that Plasmodium induced B cells survive only for a short time as short-lived memory B cells. This may provide some explanation to the authors' query.

We included data on ex-vivo expression of CD80, a marker of T dependent B cell responses, on total B cells, resting memory and activated memory B cells as a supplementary figure (Figure 5—figure supplement 2) in the revised manuscript. CD80 expression did not differ significantly between P and NP subjects at any time point, although a trend of higher expression in activated memory B cells at T6 and T7 in the protected subjects was noted.

We also analyzed the data for differences in the frequencies of dead cells between DFD vs. STD regimens at T5 (6 days post dose 3) based on Live/Dead aqua staining (Aqua^+^/Aqua^-^) on T cells and B cells as shown in Author response image 1. Mean frequencies of dead cells within the T cell compartment were 5.78 ± 3.6 for DFD and 6.3 ± 3.9 for STD regimen. For the B cells, the mean frequencies of dead cells were 8.6 ± 5.0 for the DFD and 9.2 ± 4.9 STD arms. The differences in the frequencies dead cells for either T or B cells were not significant between the two regimens. We mentioned this information in the Discussion section.

**Author response image 1. respfig1:** Frequencies of dead cells in T cell and B cell compartments at T5 (6 days post dose 3). Dead cells were gated from total CD3^+^ and CD20^+^ cells based live dead aqua staining as CD3+aqua+and CD 20+aqua+ cells. Scatter dot plots showing frequencies of dead cells for T cells (top right) and B cell (bottom right) betweeen DFD vs. STD regimen.

The labeling of the figures should be improved, particularly with regard to the time of PBMC sampling versus vaccinations. The authors need to indicate by arrows on the X axis the timing of vaccination vis-a-vis samplings. This will frame the scheduling indicated by T0, T1, etc. Otherwise, one needs to constantly consult the text for clarification.

We revised the figures and included information on the timing of vaccination, PBMC sampling and *Plasmodium* challenge.

The writing in general is a bit confusing. For example, (1) In the Discussion – "A major objective of.……was to determine if one regimen, DFD or STD, was superior…" The results of the superiority of DFD over STD in protection was published previously. Please reword the sentence; (2) "…that differences in immunity in the DFD regimen arose…" (3) "…B cell maturation subsets…"; (4) previously conducted studies are not referenced.

We revised the manuscript to address the concerns about the writing style.

In the Abstract the authors state that B cells are rescued by the DFD immunization regimen. It is true that B cells producing IgG responses declined (Figure 3). Are B cell numbers also lost? I do not recall any studies where the authors show apoptosis or some other form of B cell death induced by the STD immunization regimen.

We did not include apoptosis markers in the B cell flow cytometry staining panel. Frequencies of dead B cells were estimated from the flow cytometry gating using live/Dead Aqua staining as shown in Figure 1 in the response to comment 6. As discussed above, we did not find significant differences in the frequencies of dead B cells comparing the DFD vs. STD regimens. We are not sure about whether the total B cell numbers decreased, as we don’t have the information for absolute B cell numbers for the DFD and. STD regimens. However, the frequencies of B cells did not change between DFD vs. STD regimens. Our data suggests that the DFD regimen may result in a qualitative improvement within the B cell compartment as compared to the STD regimens.

Reviewer #2:[…]1) It is unclear from Figure 2 how flow gates were set; ideally this could be based on FMO controls, which should be shown in the gating panel, but I could not find any mention of how gates were determined in the Materials and methods. The cutoff for ICOS, IL21, and Ki67 looks rather arbitrary, and the Ki67 stain did not appear to provide good separation based on the (very small) example plot.

We revised the gating strategy figures for IL-21, ICOS and Ki67 (now Figure 3 in the revised manuscript). We included the dot plots for fluorescence minus one (FMO) staining controls to clarify the gating issues. The gating strategy write-up was revised in the Materials and methods to include the gating control information. In addition, we included a new supplementary figure (Figure 2—figure supplement 1) showing the gating strategy that was used to identify total pTfh, CSP-specific CD4, and CSP-specific pTfh cells.

2) No flow cytometry plots for B cell subsets are provided. These are critical data and should be shown (with Figure 4 or as Supplemental). This is of particular concern given that the staining for Ki67 on T cells (Figure 2C) was not very convincing, and Ki67^+^ B cells comprise the majority of the most predictive parameters in Table 1.

A detailed gating strategy for the identification of B cell subsets and Ki67 expression on various B cell subsets is included as Figure 5—figure supplement 1 in the revised manuscript. The definition of B cell subsets based on the markers used, is further clarified in the Materials and methods and Results sections.

3) What do the error bars indicate in Figure 1-4 (SD, SEM, other)? In some cases the error bars are quite short relative to the difference between P and NP values, yet the difference is not significant; while in the puzzling case of Figure 2F (T2) the difference IS statistically significant yet the values are almost the same and the bars widely overlap.

All error bars in the line graphs indicate the standard error of mean (SEM). The data for Figure 2F at T2 was rechecked and corrected as we found an error in the data for the protected subjects. The difference between P and NP subjects at T2 is significant with a p value of <0.01.

4) The definition of Tfh is somewhat non-conventional and only partially justified (i.e. in Discussion re: PD-1).

We identified the phenotype of CSP-specific pTfh as CD4^+^CD40L+ cells expressing CD45RO and CXCR5 and also investigated their cytokine profile for IL-21 and IFNγ. We did not include markers associated with pTfh heterogeneity such as PD1, CD38, CXCR3 etc. in the 15-color flow cytometry panel that was used to study the Ag.pTfh. Since we were focusing on a less frequent pTfh population within the CSP-specific CD4 T cells compartment, we rationalized that we needed to maintain a broad(er) definition of pTfh cells as we studied their antigen-specific function with the goal of finding novel determinants of malaria vaccine-induced immune responses. In previous studies, we saw that some of the phenotypic markers used for pTfh identity such as PD1 and CD38 are also influenced by age or other infectious diseases (De Armas et al., 2017). Moreover, in an influenza vaccine study, we found that activated pTfh cells (HLA-DR+CD38+) expressing PD1 appear to be detrimental to the antigen-specific pTfh responses (Pallikkuth et al., 2019). This has been discussed in the Discussion section.

5) The reader has to work rather hard to grasp what comparisons were statistically significant. The differences from T0 to other timepoints stand out (but are not the most interesting result). This overshadows the p-values comparing P and NP. Only the latter are mentioned in the Figure Legend. (applies to Figures 1-3).

We revised all the figures and included information on the timing of vaccination, blood sampling and *Plasmodium* challenge. In order to address the differences between P and NP subjects at each timepoint, we revised the figures by including connecting lines between timepoints in which differences between P vs. NP are significant.

6) The machine learning portions of the paper (rationale, findings) are nicely explained in the Discussion, but are presented in a very confusing manner in the Results. I did not understand the point of this analysis until the Discussion.

The machine learning section of the Results section has been re-organized and significant parts of it have been re-written. One paragraph from the machine learning section in the original draft was moved into a new section of the Results (“Identifying vaccine-induced immune responses in RTS,S”) to make the section more clear. We have revised Results section and Materials and methods section to highlight the changes.

7) Figure 5 was not helpful to this reviewer and could be considered for omission; conversely Figure S1 showing the trial schematic with definition of timepoints T0-T7 was very helpful and should be considered for inclusion in the main text.

As suggested by the reviewer, we have moved the supplementary figure 1 in the original submission showing study schematics as a main figure (Figure 1) and Figure 5 has been included as Figure 6—figure supplement 1 in the revised manuscript.

8) Was any analysis of how Tfh correlate with B cells in individuals performed? This would be of interest but appears to be missing.

We carried out a correlation analysis between all pTfh and all memory B cell parameters at each timepoint and did not find any direct correlations between pTfh parameters and B cell measures. This observation combined with the early pTfh responses in protected subjects, suggest that initial Tfh cell priming may be required for the subsequent development of a qualitatively superior Tfh-induced B cell response induced by RTS,S/AS01B vaccine. The actual timing and magnitude of pTfh response that need to be elicited in order to influence the quality of the later B cell responses and improved protection in the DFD regimen need further investigations. We have discussed the results and implications in the Discussion.

9) The paper would benefit from a discussion of the implications of these findings on RTS,S vaccination in real-world endemic settings, where many children would inevitably be exposed (often repeatedly) to malaria antigens during the "suboptimal" 2-6 month window following priming (see Obeng-Adjei et al., 2015).

While we did not conduct detailed assessments of Tfh and B cell responses following RTS,S vaccination in the field, we are currently in the process of developing a study that will address the role of select T and B cell subpopulations, such as atypical memory B cells, Tfh and gd T cells, in the context of RTSS immunization and immune hypo-responsiveness in Kenyan adults. We have included this information in the Discussion.

Reviewer #3:[…]1) What is the qualitative difference between the protected versus non-protected subjects in terms of the cTfh levels or its functions?

In the revised manuscript, we show the ex-vivo frequencies of total pTfh (CD45RO^+^CD27^+^CXCR5^+^) cells in Figure 2A; ICOS expression on ex-vivo total pTfh is mentioned in the Results section. Our data show that frequencies of total pTfh were significantly different between P and NP subjects at T3, T4, T6 and T7 and the ICOS expression on total pTfh was significantly higher in P subjects at T6 and T7 compared to NP subjects. We also analyzed the IL-21 expression in total pTfh after 12 hrs of CSP stimulation, and similar to the observed increase in IL-21 production in the Ag.pTfh cells post-vaccination, there was a trend of higher IL-21 expression in total pTfh cells in P subjects post-vaccination (Author response image 2). Taken together, we consider that pTfh are quantitatively and qualitatively superior in P subjects as compared to NP subjects. We have included this information in the Results section.

**Author response image 2. respfig2:** Trend of higher frequencies of total pTfh cells expressing IL-21 after vaccination in P subjects. Il-21^+^ total pTfh cells after CSP stimulation by flow cytometry. IL-21+ pTfh cells were identified as CD4^+^CD45RO^+^CD27^+^CXCR5 cells.

2) The authors indicate that expansion of IL-12 and ICOS^+^ Ki67^+^ cells is indicative of vaccine-induced enhancement; however, this happens in both the arms. It did not look like this is happening only in the DFD arm. Results are written as if they are significantly different.

Our conclusions are based on the increase in IL-21, ICOS, and Ki67 comparing baseline to post-vaccine timepoints in protected subjects, indicating a vaccine-induced enhancement of these three immune parameters in the CSP-specific pTfh cells. Non-protected subjects did not show an increase in these markers. Moreover, in the DFD arm there was significantly higher IL-21 and ICOS expression at T7 compared to the STD arm, which further supports a regimen-induced enhancement of the pTfh function.

3) Do the authors have any hypothesis for the enhancement of B and Tfh cell response in DFD group compared to the STD regimen?

One possible hypothesis is that early Tfh response induces by the initial vaccine doses result in the formation of a strong high affinity memory B cell pool specific to CSP antigen and the delayed fractional dose lead to expansion and differentiation of the preformed memory B cells to Ab secreting cells. In the case of STD vaccine regimen, early administration of the booster dose was detrimental to the pre-formed memory B cell expansion and differentiation. To support our hypothesis, our data integration analysis showed that nearly 30% of good “early” responders in the STD arm became non-protected after the 3^rd^ dose, indicating that dose 3 within a 0, 1, 2 month dose schedule may have an adverse effect on an otherwise promising immune response generated by dose 1 and 2.

4) Did they do an antigen-specific IgG subclass response to see whether DFM has wide range of antibody responses compare to STD regimen?

We analyzed the total IgG levels in PBMC culture supernatants after 5 days of stimulation with malaria antigens, such as full-length CSP (PF-CSP), C-term CSP (PF-16), and repeat region (R32LR). Responses to each of these antigens were included as a variable in the data integration analysis and none of the IgG responses were selected by the model as being associated with protection or regimen difference. We included this information in the revised manuscript as a supplementary figure (Figure 5—figure supplement 5). Moreover, the serum IgG response specificity/isotype response from this trial was published previously by our collaborators (Chaudhury et al., 2017).

5) It is also surprising to see that non-Tfh component are not looked t properly such as IFN-γ, TNF-α, and IL-2 levels. What is the association of these cells with antigen specific B cell response between P versus NP individuals?

We have analyzed non-Tfh compartment and IFNγ responses, while IL-2 or TNF in the Tfh compartment were not analyzed in this study. Non-Tfh and IFNγ data were included as variables in the data integration analysis and neither was selected as a variable associated with protection or regimen difference in the data integration analysis. We included this information in the Discussion.

6) In the Figure 7 it is not very clear that what predicts the protection and what is not predicting the protection.

We have included the information related parameters predictive of protection in this model as a new Supplementary file 1B.

[Editors' note: further revisions were suggested prior to acceptance, as described below.]

1) In their response to a comment on non-pTfh cells, the authors rely on negative data (not shown) derived from the integration analysis. Despite the outcome of this analysis, it might be useful and most informative to assess the non-pTfh cells alongside the pTfh cells. In the event that the PBMCs are no longer available, the authors need to address these potentially crucial components as possible mechanisms for the superior outcome in DFD regimen of immunization. As an example, previously published works have shown an association between IL-2^+^ and IFNγ^+^ CD4 T cells and antibody responses in RTS,S vaccinated subjects. In the absence of results, including these points would improve the tenor of the Discussion.

We have now included the non-pTfh data in the Results section of the manuscript as a supplementary figure (Figure 2—figure supplement 2).

2) Please revisit the legends and confirm whether they indicate that the error bars signify SEM.

The information about the error bars signifying SEM is included in all figure legends.

3) Figure 3. Unlike IL-21 and ICOS, the expression of Ki67 was rather transitory and this point needs to be acknowledged.

We have now mentioned the transitory nature of the Ki67 expression in the Results

4) A title should be more reflective of the key findings.

We have changed the manuscript title to be more reflective of our findings. The new title is “A delayed fractionated dose RTS,S AS01 vaccine regimen mediates protection via improved B cell responses”.

5) In the Abstract, there should be a clearer pivot from data that is previously published (trial results) to data presented therein.

Abstract has been revised.

6) Subheading “CSP-specific B cell responses emerge in protected subjects after the second dose and higher regimen specific memory B cell response in DFD arm” makes no sense grammatically.

We have revised the subheading.

7) In the Discussion, the authors state that "This is the first study demonstrating a protective role of CSP-specific pTfh in RTS,S/AS01 vaccination". Please tone down this statement as the results do not show a protective role of pTfh cells, they simply show an association.

We have revised the Discussion section and the suggested statement has been removed.

8) Discussion section, final two sentence in the penultimate paragraph should could be removed, as they are unnecessary.

Deleted as suggested.

9) Results section requires extensive editing in that a rationale for choosing to ask a certain question or perform a particular experiment needs to be clearly stated. It is insufficient to state, "We next investigated…”.

We have extensively revised the Results section to indicate the rationale/question related to each analysis performed.

10) Discussion paragraph six is very repetitive to early wording in the Results.

The Discussion has been revised.

11) The Discussion section on the whole still appears largely as a rehash of the Results section. It fails to provide a cogent picture or a hypothesis that would suggest possible interactions that may be occurring between the pThf cells (and possibly non-pTfh CD4 T cells) and the various B cells subsets and which bring about this enhanced protection seen in subjects receiving DFD regiment.

We have revised the Discussion extensively to discuss the hypothesis that early Tfh response induced by the initial vaccine dose results in the formation of a strong high affinity memory B cell pool specific to CSP antigen, and the DFD leads to expansion and differentiation of the preformed memory B cells to Ab secreting cells.

12) Along these lines, the authors need to relate their finding to a larger picture of RTS,S vaccination in African countries. Specifically, the paper would benefit from a discussion of the implications of these findings on RTS,S vaccination in real-world endemic settings, where many children would inevitably be exposed (often repeatedly) to malaria antigens during the "suboptimal" 2-6 month window following priming. The authors are not requested to provide field data, which they obviously cannot; however if a DFD regimen were deployed in Africa, many children would get a natural malaria infection that might have the same negative impact as the STD dose #3 and this is worth discussing.

We have revised the Discussion to relate our findings to the field. We have updated the section in the Discussion reflecting on the induction of Tfh cells by RTS,S/AS01E vaccination in African children, but balanced by findings that malaria infection can induce dysfunctional Tfh phenotypes. In future DFD trials in the field, the induction of Tfh cells should be investigated.